# Enhancement of CO$_2$ Reforming of CH$_4$ Reaction Using Ni,Pd,Pt/Mg$_{1-x}$Ce$_x$$^{4+}$O and Ni/Mg$_{1-x}$Ce$_x$$^{4+}$O Catalysts

**Faris A. J. Al-Doghachi [1,\*], Ali F. A. Jassim [2] and Yun Hin Taufiq-Yap [3,4,\*]**

[1]   Department of Chemistry, Faculty of Science, University of Basrah, Basrah 61004, Iraq
[2]   Department of Chemical Engineering, Faculty of Engineering, Universiti Putra Malaysia, 43400 UPM, Serdang, Selangor 40000, Malaysia; ali_faris2015@hotmail.com
[3]   Chancellery Office, University Malaysia Sabah, 88400, Kota Kinabalu, Sabah 88000, Malaysia
[4]   Catalysis Science and Technology Research Centre, Faculty of Science, Universiti Putra Malaysia, 43400 UPM, Serdang, Selangor 40000, Malaysia
\*   Correspondence: farisj63@gmail.com (F.A.J.A.-D.); taufiq@upm.edu.my (Y.H.T.-Y.); Tel.: +60-111-1658-064 (F.A.J.A.-D.); +60-389-466-809 (Y.H.T.-Y.)

**Abstract:**   Catalysts Ni/Mg$_{1-x}$Ce$_x$$^{4+}$O and Ni,Pd,Pt/Mg$_{1-x}$Ce$_x$$^{4+}$O were developed using the co-precipitation–impregnation methods. Catalyst characterization took place using XRD, H$_2$-TPR, XRF, XPS, Brunauer–Emmett–Teller (BET), TGA TEM, and FE-SEM. Testing the catalysts for the dry reforming of CH$_4$ took place at temperatures of 700–900 °C. Findings from this study revealed a higher CH$_4$ and CO$_2$ conversion using the tri-metallic Ni,Pd,Pt/Mg$_{1-x}$Ce$_x$$^{4+}$O catalyst in comparison with Ni monometallic systems in the whole temperature ranges. The catalyst Ni,Pd,Pt/Mg$_{0.85}$Ce$^{4+}$$_{0.15}$O also reported an elevated activity level (CH$_4$; 78%, and CO$_2$; 90%) and an outstanding stability. Carbon deposition on spent catalysts was analyzed using TEM and Temperature programmed oxidation-mass spectroscopy (TPO-MS) following 200 h under an oxygen stream. The TEM and TPO-MS analysis results indicated a better anti-coking activity of the reduced catalyst along with a minimal concentration of platinum and palladium metals.

**Keywords:** biogas; dry reforming; catalyst deactivation; syngas; H$_2$ production

## 1. Introduction

Greenhouse gases (CH$_4$ and CO$_2$) production from anaerobic biomass digestion has been utilized as fuel for the production of power and heat. Furthermore, it has been implemented as a renewable source of carbon in the syngas (CO and H$_2$) production for industrial reactant materials through a reaction that is both economic and environmentally friendly. The dry reforming method is considered a surrogate method in which carbon dioxide is used as an oxidant (Equation (1)).

$$CH_4 + CO_2 \rightarrow 2CO + 2H_2 \qquad (1)$$

Among the crucial raw materials is syngas, which can adequately be transformed to oxygenated ultra-clean fuels (methanol, dimethyl ether gasoil, and gasoline) via the utilization of Fischer–Tropsch reactions [1]. The reaction of reverse water-gas shift (RWGS) has been shown to affect the reaction equilibrium during the CO and H$_2$ production from CO$_2$ and CH$_4$ (Equations (1) and (2)) by reducing the ratio of H$_2$/CO.

$$CO_2 + H_2 \rightarrow CO + H_2O \qquad (2)$$

The reaction of dry reforming has also been linked to other reactions, e.g., the methane decomposition reaction (Equation (3)), and the disproportionation reaction (Equation (4)).

$$CH_4 \rightarrow C + 2H_2 \tag{3}$$

$$2CO \rightarrow C + CO_2 \tag{4}$$

Al-Doghachi et al. [2] reported a direct relationship between the decomposition of $CH_4$ (Equation (3)) and CO disproportionation (Equation (4)) to the carbon formation on the catalyst. The same study reported that when the reaction temperature escalated from 550–650 °C, the likelihood for the formation of carbon in preference to DRM was noted. Hence, the selection of the catalyst plays an imperative function in curbing the coke deposition and the enhancement of the DRM reaction. Moreover, a reduction/elimination of the carbon deposition was observed through the catalyst support basicity, attributed to an enhancement in the support's basicity leading to the chemisorption of the catalyst to carbon dioxide in the $CO_2$ reforming of $CH_4$ [3]. Hence, this gas reacts with C to form CO (Equation (5)).

$$CO_2 + C \rightarrow 2CO \quad \Delta H^{\circ}_{298} = +172 \, KJ/mol \tag{5}$$

When designing anti-carbon deposition catalysts, it is imperative to consider the two significantly remarkable factors, first of which is the carbon deposition that can only occur upon a higher size of the metal as compared to the critical size. Another component is the carbon formation that is preferred by an acidic support. Thus, for reducing the coke formation, the metal cluster size is required to be of a size that is lesser than the critical size required for the carbon formation with a reduction in the acidity of the support. The most efficient way reported to minimize the acidity of the support is by utilizing metal oxides that are basic (i.e., alkaline earth metal oxides), and can effectively function as a support [3]. Selecting MgO as a support was attributed to its characteristics such as preventing acidity and controlling the size of the main catalyst (Ni particles), by using minor NiO concentrations (~1%) to inhibit the deposition carbon as well as sintering. Other advantages of using MgO as support lie in its high thermal stability at a high melting point (2850 °C) and its low cost of production [2]. Ni-MgO-$Al_2O_3$ nanocatalysts were prepared by Akbari et al. [4] with varying cerium contents using the impregnation and co-precipitation techniques. Findings from the study demonstrated high coke resistance and high $CO_2$ and $CH_4$ conversion. Jin et al. [5] improved the performance of the ALD-prepared Ni/$Al_2O_3$ catalyst by the addition of $CeO_2$. In their study, the addition of $CeO_2$ resulted in a halt in the coke deposition and an increase in the stability of catalysts that were nickel-based.

Recently, the use of bi and tri-metallic catalysts was shown to result in a higher catalytic selectivity, activity, and stability, which can be supported by findings from studies on the bi-metallic catalyst Co,Ni/$CeO_2$ [6] and Pt,Ni/MgO [7] that reported a good $CO_2$ and $CH_4$ conversion rate in the DRM reaction. Several studies comparing the DRM yield of monometallic, bimetallic, and tri-metallic catalysts reported an elevated level of conversion in the tri-metallic catalysts as compared to the monometallic catalysts [8]. Khoja et al. [9] studied the performance of Ni-based catalysts with $Al_2O_3$ and modified $Al_2O_3$ support and found an enhancement in the catalyst stability and activity in the Al-La and Al-Mg supports in comparison to the unmodified counterpart. Regardless of the extensive research on $ZrO_2$, MgO, and $Ce_2O_3$ as catalytic promoters, there is a little investigation regarding the utilization of $Ce^{4+}$ as a promoter of Ni/MgO for DRM [10].

The goal of this study was to examine the role of the Pt and Pd metals on Ni/$Mg_{1-x}Ce_xO$ catalysts, which was reported to provide electronic density on the main catalyst Ni and hence prevent the catalyst sintering and carbon deposition and improve the selectivity and $CO_2$/$CH_4$ conversion. To further investigate the effect of the metals on the catalytic performance during DRM reaction, the selectivity, stability, and activity of the Ni,Pd,Pt/$Mg_{1-x}Ce_xO$ catalysts, and Ni/$Mg_{1-x}Ce_xO$ catalysts were compared. The catalysts were characterized by XRD, XRF, XPS, TEM, $H_2$-TPR, FE-SEM Brunauer–Emmett–Teller (BET), and TGA. The study has also evaluated the stability of the catalysts and tested the role of $CH_4$ and

$CO_2$ concentrations, the catalysts' concentration, the prepared catalysts conversion temperature, and the efficacy of the catalyst in the process of dry reforming. Temperature programmed oxidation-mass spectroscopy (TPO-MS) and TEM analysis were carried out to examine the carbon deposition on the spent catalysts following 200 h of using the catalyst. Ultimately, the research investigated the enhancement of the methane conversion through passing oxygen gas stream (1.25%) across the reaction.

## 2. Results and Discussion

### 2.1. Characterization of Catalysts

### 2.1.1. Patterns of XRD

Figure 1a–d shows the XRD results for the catalysts with MgO and $CeO_2$ components. The diffraction peaks at 2θ = 37.0 (111), 42.9 (200), 62.2 (220), 74.4 (310), and 79.1° (222) were ascribed to the cubic MgO form (JCPDS file no.: 00-002-1207). Concurrently, recordings at 2θ = 28.4 (111), 33.2 (200), 47.3 (220), 56.2 (311), 59.3 (222), 69.3 (400), 76.5 (330), and 79.1° (411) were assigned to ceria in the cubic form (JCPDS file no.: 00–034–0394) [11]. Peaks obtained at 2θ = 47.5, 56.3, 62.2, 69.4, 69.8, and 79.1° may be due to the catalyst complex in cubic form. In all the patterns (Figure 1A), no peaks of diffraction were recorded for the catalyst Ni (1%) as a result of the trace amount of Ni element and the inability of the machine to sense elements lesser than 5%. Whereas, all patterns in Figure 1B displayed small lines at 2θ 12–27° that were probably related to Pd (1%) main catalyst [10]. This finding agrees with Al-Doghachi et al. results [12,13]. Using Debye–Scherrer's formula and through the soaring diffraction peak in the XRD, the average crystalline size was identified (Table 1). Findings reported that the crystal size was directly proportionate to the elevated $CeO_2$ amount in the catalysts, which can be linked to the role of the elements Ni, Pd, and Pt, that persisted on the surface and hence halted the MgO crystallites growth. The measurements of the crystal sizes for $Ni,Pd,Pt/Mg_{0.85}Ce^{+4}_{0.15}O$, $Ni,Pd,Pt/Mg_{0.93}Ce^{+4}_{0.07}O$, $Ni,Pd,Pt/Mg_{0.97}Ce^{+4}_{0.03}O$, and Ni,Pd,Pt/MgO, were 52, 52, 51, and 42 nm respectively. However, the crystal size measurement of Ni/MgO, $Ni/Mg_{0.97}Ce^{4+}_{0.03}O$, $Ni/Mg_{0.93}Ce^{4+}_{0.07}O$, and $Ni/Mg_{0.85}Ce^{4+}_{0.15}O$ were 39, 52, 52, and 53 nm, respectively. A cubic form was recorded for the crystal system for all samples due to the cubic particles on the inside of the catalyst [14]. Through the TEM figures, it was shown that most of the catalyst crystals were cubic supporting the XRD findings.

**Table 1.** XRD, TEM, and XRF results for the measurement of particle sizes.

| Catalysts | TEM (nm) | [a] Crystal Size (D) Debye-Sherrer's Equation (nm) | XRF | | | |
|---|---|---|---|---|---|---|
| | | | Ni% | Pd% | Pt% | Mg & Ce% |
| Ni/MgO | 44 | 39 | 0.91 | | | 98.7 |
| $Ni/Mg_{0.97}Ce^{4+}_{0.03}O$ | 56 | 52 | 0.95 | | | 98.3 |
| $Ni/Mg_{0.93}Ce^{4+}_{0.07}O$ | 58 | 52 | 1.03 | | | 98.6 |
| $Ni/Mg_{0.85}Ce^{4+}_{0.15}O$ | 59 | 53 | 0.93 | | | 98.5 |
| Ni,Pd,Pt/MgO | 53 | 42 | 0.96 | 1.09 | 1.15 | 96.2 |
| $Ni,Pd,Pt/Mg_{0.97}Ce^{4+}_{0.03}O$ | 54 | 51 | 1.03 | 1.24 | 1.03 | 96.0 |
| $Ni,Pd,Pt/Mg_{0.93}Ce^{4+}_{0.07}O$ | 55 | 52 | 0.93 | 1.04 | 1.15 | 96.4 |
| $Ni,Pd,Pt/Mg_{0.85}Ce^{4+}_{0.15}O$ | 57 | 52 | 0.96 | 1.22 | 0.98 | 95.8 |

[a] Determined by the Debye–Scherrer's equation of the Mg (200) plane of XR.

XRF method has been implemented for the analysis of elements in all the catalyst components. Table 1 illustrates the Ni, Pd, and Pt percentages. In the co-precipitation method, the incomplete Mg and Ce metal precursors precipitation led to percentages that were marginally more than 1. This affected the results slightly [2].

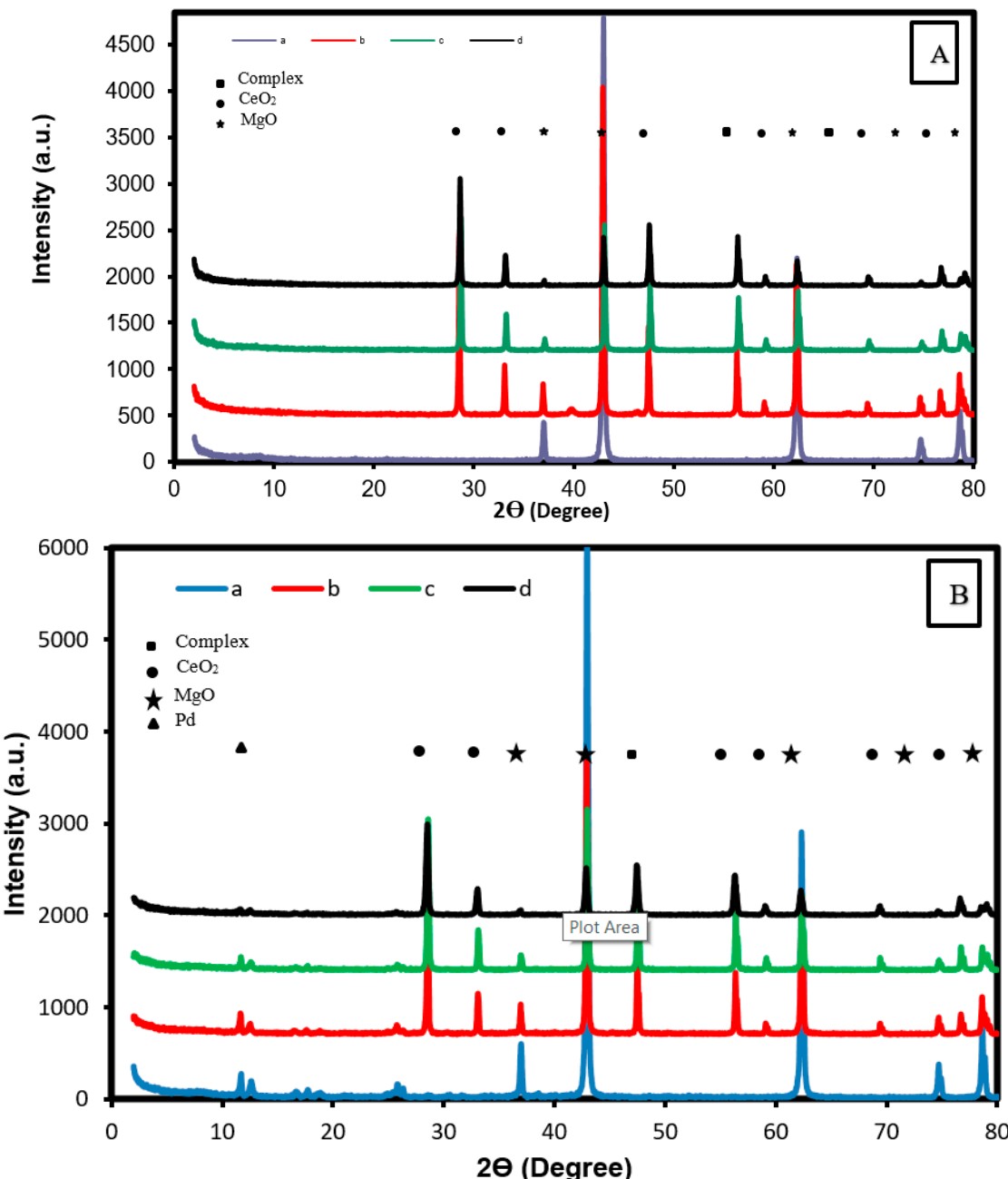

**Figure 1.** XRD patterns of the (**A**) Ni/Mg$_{1-x}$Ce$^{4+}$$_x$O and (**B**) Ni,Pd,Pt/Mg$_{1-x}$Ce$^{4+}$$_x$O catalysts, were (x = a (0.00), b (0.03), c (0.07), and d (0.15)).

### 2.1.2. XPS Analysis

The analysis of XPS was implemented to investigate the surface composition of the reduced catalyst Ni/Mg$_{0.85}$Ce$^{4+}$$_{0.15}$O. An XPS examination of the surface of the catalysts with 3–12 nm displayed the photoelectron signals emittance from O1s, Mg2p, Ni2p, and Ce3d Figure 2A–E. The photoelectron signal, O1s, displayed the peaks for Mg–O, Ce–O, and Ni–O, that were deconvoluted at binding energies 530, 529.2, and 529.1 eV, respectively, as shown in Figure 2B. The narrow scan of the XPS spectra for the Mg2p region of the nanocatalyst recorded one peak at 47 eV binding energy (Figure 2C); whereas, the other peaks located at the binding energies around 854, 860, 871, and 878 eV were assigned to the Ni2p. Finally, the narrow scan of XPS spectra for the Ce3d region of the nanocatalyst deconvoluted into complex peaks and displayed the highest photoelectron signal intensity in the

binding energies 917, 905, 900, 896, 887, and 880 eV, in comparison with the other peaks. The Mg2p, Ce2p, and Ni2p narrow scan demonstrated a combination of Mg–O, Ce–O, and Ni–O in the oxide species composition of these metals, respectively [15].

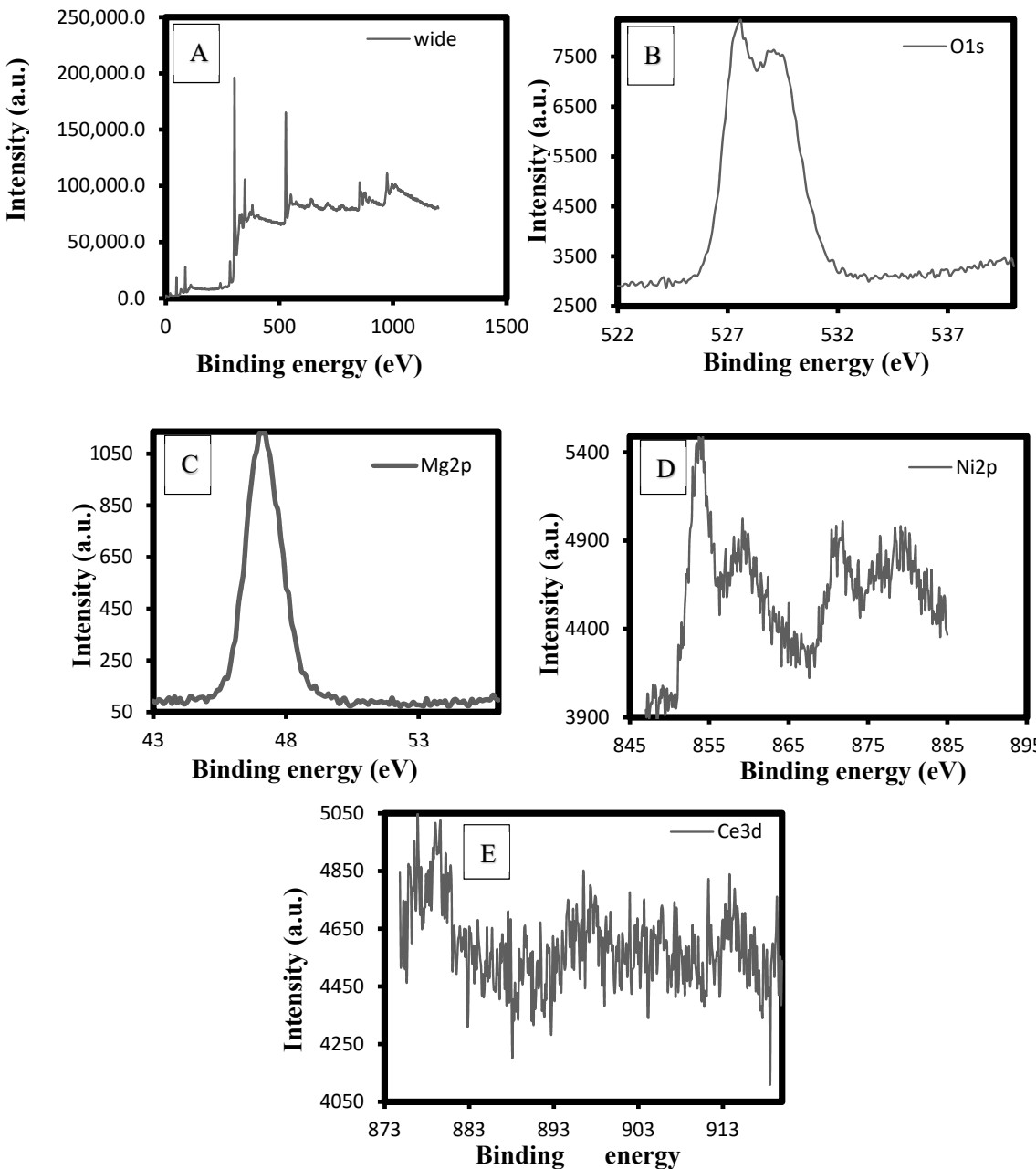

**Figure 2.** Narrow XPS scans of the Ni/Mg$_{0.97}$Ce$^{4+}$$_{0.03}$O catalyst. (**A**) Wide; (**B**) O1s; (**C**) Mg2p; (**D**) Ni2p; (**E**) Ce3d.

### 2.1.3. TEM and FE-SEM Characterization

Figure 3A–H demonstrates the distribution and morphology images of the synthesized catalysts by TEM (shape and size). The crystals were characterized by analysing the cubic structure and particle sizes using TEM. Figure 4A–H illustrates the catalysts topology images that were obtained by FE-SEM and were supported by the TEM analysis. Regular-shaped particles were observed by the catalysts Ni,Pd,Pt/MgO, Ni,Pd,Pt/Mg$_{0.97}$Ce$^{4+}$$_{0.03}$O, Ni,Pd,Pt/Mg$_{0.93}$Ce$^{4+}$$_{0.07}$O, and Ni,Pd,Pt/Mg$_{0.85}$Ce$^{4+}$$_{0.15}$O [16]. Meanwhile, due to the low concentration (1%) of the metals Ni, Pd, and Pt, the atoms on the surface

of the catalyst were further from one another with no agglomeration, which is required by the DRM reaction. Figure 3A–H shows a 2-D cubic texture devoted to the catalyst [17]. The catalyst pores were of uniform size (~18 nm), which concurred with the findings from the BET as shown in Table 2. Several particles of Pt, Pd, and Ni were loaded on the outer surface of the $Mg_{0.85}Ce_{0.15}O$ support uniformly, which markedly differed from the crystalline sites inside the porous structure. The discrepancy in the size of the metal particle can be attributed to the regulated crystals growth inside the narrowly distributed channels. This led to reduced Ni particle homogeneity as compared to other catalysts. The supported Ni particle size showed an escalation as follows: $Ni,Pd,Pt/MgO < Ni,Pd,Pt/Mg_{0.97}Ce^{4+}{}_{0.03}O < Ni,Pd,Pt/Mg_{0.93}Ce^{4+}{}_{0.07}O < Ni,Pd,Pt/Mg_{0.85}Ce^{4+}{}_{0.15}O$, corresponding to the Scherrer equation results (Table 1).

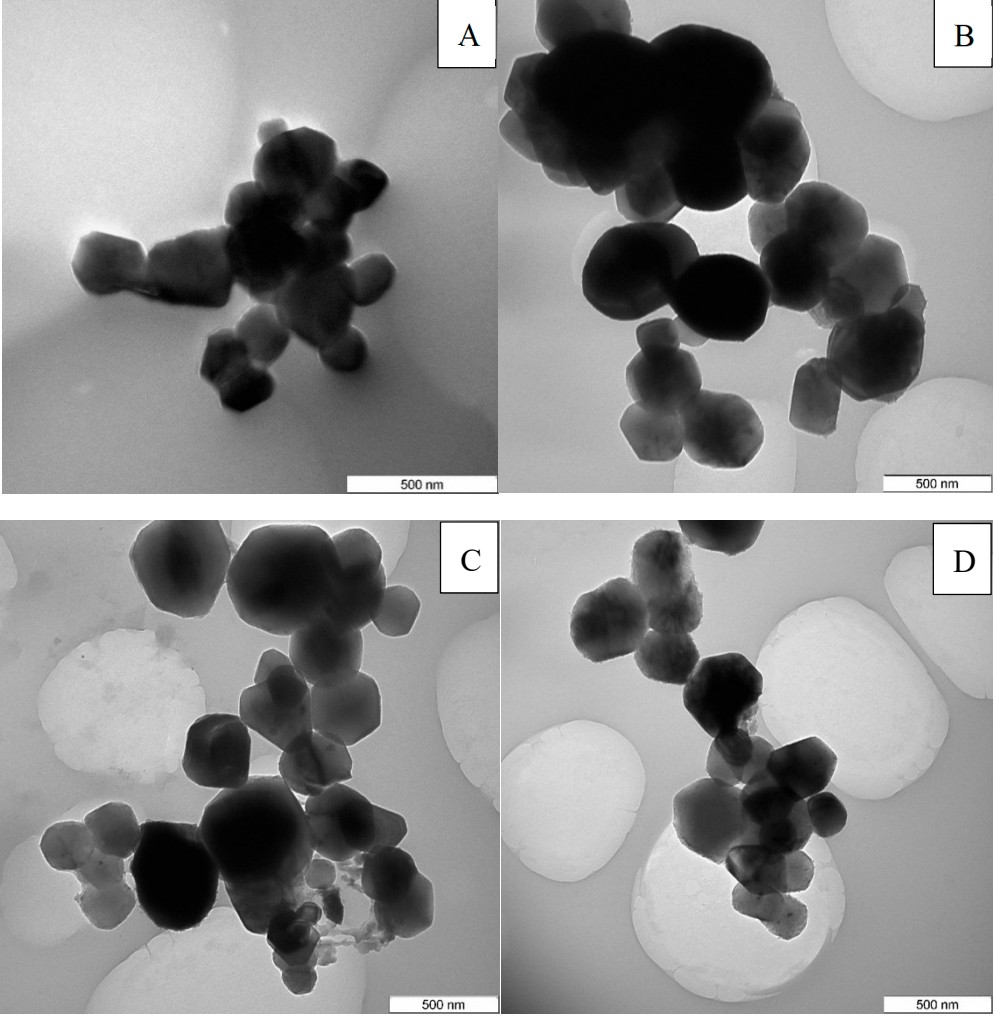

**Figure 3.** *Cont.*

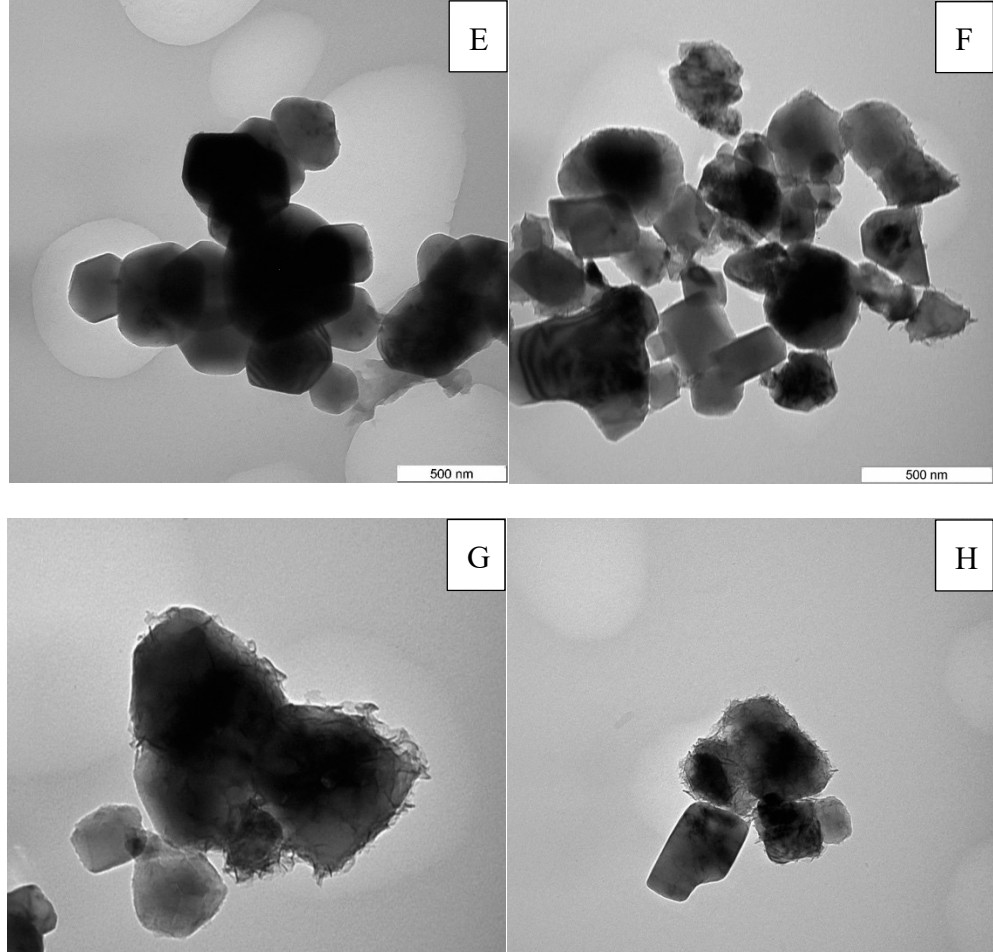

**Figure 3.** TEM figures for the catalysts: (**A**) Ni/MgO; (**B**) Ni/Mg$_{0.97}$Ce$^{4+}_{0.03}$O; (**C**) Ni/Mg$_{0.93}$Ce$^{4+}_{0.07}$O; (**D**) Ni/Mg$_{0.85}$Ce$^{4+}_{0.15}$O; (**E**) Ni,Pd,Pt/MgO; (**F**) Ni,Pd,Pt/Mg$_{0.97}$Ce$^{4+}_{0.03}$O; (**G**) Ni,Pd,Pt/Mg$_{0.93}$Ce$^{4+}_{0.07}$O; (**H**) Ni,Pd,Pt/Mg$_{0.85}$Ce$^{4+}_{0.15}$O.

**Table 2.** The fresh catalysts textural properties using Brunauer–Emmett–Teller (BET) method.

| Sample | Specific Surface Area (m$^2$/g) | Pore Volume (cm$^3$/g) | Pore Radius (Å) |
|---|---|---|---|
| Ni/MgO | 10.7 | 0.089 | 28.7 |
| Ni/Mg$_{0.97}$Ce$^{4+}_{0.03}$O | 12.1 | 0.116 | 54.8 |
| Ni/Mg$_{0.93}$Ce$^{4+}_{0.07}$O | 14.4 | 0.134 | 63.3 |
| Ni/Mg$_{0.85}$Ce$^{4+}_{0.15}$O | 17.2 | 0.185 | 68.5 |
| Ni,Pd,Pt/MgO | 10.9 | 0.122 | 44.4 |
| Ni,Pd,Pt/Mg$_{0.97}$Ce$^{4+}_{0.03}$O | 13.5 | 0.132 | 60.9 |
| Ni,Pd,Pt/Mg$_{0.93}$Ce$^{4+}_{0.07}$O | 17.1 | 0.182 | 73.3 |
| Ni,Pd,Pt/Mg$_{0.85}$Ce$^{4+}_{0.15}$O | 19.8 | 0.213 | 86.3 |

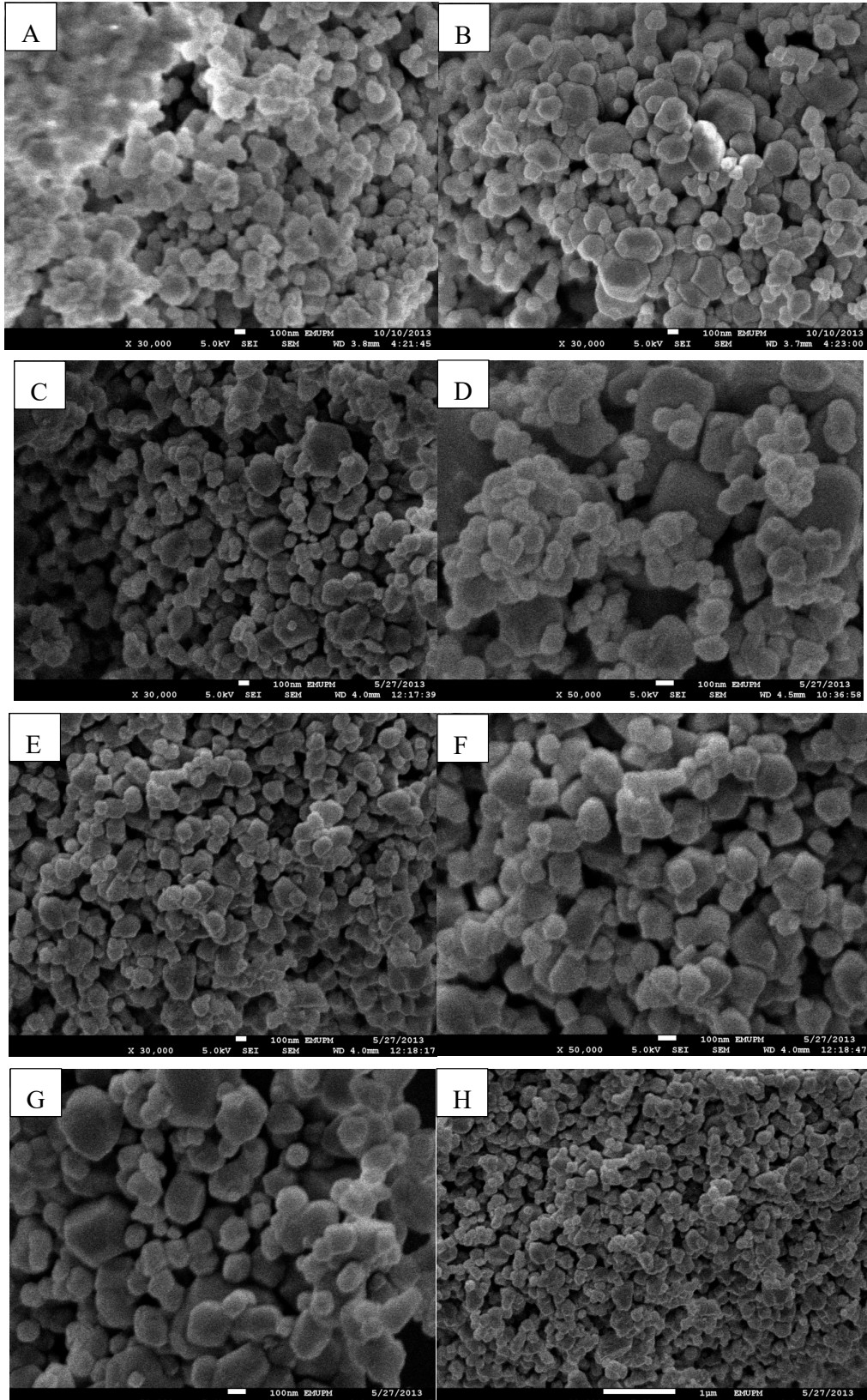

**Figure 4.** FE-SEM figures of (**A**) Ni/MgO; (**B**) Ni/Mg$_{0.97}$Ce$^{4+}_{0.03}$O; (**C**) Ni/Mg$_{0.93}$Ce$^{4+}_{0.07}$O; (**D**) Ni/Mg$_{0.85}$Ce$^{4+}_{0.15}$O; (**E**) Ni,Pd,Pt/MgO; (**F**) Ni,Pd,Pt/Mg$_{0.97}$Ce$^{4+}_{0.03}$O; (**G**) Ni,Pd,Pt/Mg$_{0.93}$Ce$^{4+}_{0.07}$O; (**H**) Ni,Pd,Pt/Mg$_{0.85}$Ce$^{4+}_{0.15}$O catalysts.

### 2.1.4. Surface are of Brunauer–Emmett–Teller (BET)

The specific BET ($S_{BET}$) surface area values are demonstrated in Table 2 along with the reduced catalyst supports pore properties. Ni/MgO catalyst had a pore radius of 28.7 Å, a pore volume of 0.089 $cm^3$/g, and a surface area of 10.7 $m^2$/g. Nonetheless, an elevation in the volume and the surface area was noticed upon the $CeO_2$ (promoter) addition. This increase may be due to the strong Ni metal and MgO-$CeO_2$ support interaction. The current findings are concurrent with the findings by earlier reports [18]. However, at a temperature of 1150 °C, the addition of $CeO_2$ curbed the surface area loss during calcination resulting in an elevation in the catalyst surface area (Table 2). The surface areas for Ni/$Mg_{0.97}Ce^{4+}_{0.03}$, Ni/$Mg_{0.93}Ce^{4+}_{0.07}O$, and Ni/$Mg_{0.85}Ce^{4+}_{0.15}O$ were 12.1, 14.4, and 17.2 $m^2$/g respectively, while the 10.9, 13.5, 17.1, and 19.8 $m^2$/g values corresponded to the surface areas of Ni,Pd,Pt/MgO, Ni,Pd,Pt/$Mg_{0.97}Ce^{4+}_{0.03}O$, Ni,Pd,Pt/$Mg_{0.93}Ce^{4+}_{0.07}O$, and Ni,Pd,Pt/$Mg_{0.85}Ce^{4+}_{0.15}O$, respectively.

Findings revealed that when the amount of Ce increases, the surface area increases. Furthermore, the $CeO_2$ concentration level affected all the catalysts pore radius. A pore radius of 28.7, 54.8, 63.3, and 68.5 Å was recorded for Ni/MgO, Ni/$Mg_{0.97}Ce^{4+}_{0.03}O$, Ni/$Mg_{0.93}Ce^{4+}_{0.07}O$, and Ni/$Mg_{0.85}Ce^{4+}_{0.015}O$, respectively; whereas, the recordings at 44.4, 60.9, 73.3, and 86.3 Å corresponded to Ni,Pd,Pt/MgO, Ni,Pd,Pt/$Mg_{0.97}Ce^{4+}_{0.03}O$, Ni,Pd,Pt/$Mg_{0.93}Ce^{4+}_{0.07}O$, and Ni,Pd,Pt/$Mg_{0.85}Ce^{4+}_{0.015}O$, respectively. Upon adding cerium, a slight elevation was observed in the pore volume to 0.089, 0.116, 0.134, and 0.185 $cm^3$/g for Ni/MgO Ni/$Mg_{0.97}Ce^{4+}_{0.03}O$, Ni/$Mg_{0.93}Ce^{4+}_{0.07}O$, and Ni/$Mg_{0.85}Ce^{4+}_{0.015}O$, respectively. Likewise, the recordings for Ni,Pd,Pt/MgO, Ni,Pd,Pt/$Mg_{0.97}Ce^{4+}_{0.03}O$, Ni,Pd,Pt/$Mg_{0.93}Ce^{4+}_{0.07}O$, and Ni,Pd,Pt/$Mg_{0.85}Ce^{4+}_{0.015}O$ catalysts were observed at 0.122, 0.132, 0.182, and 0.213 $cm^3$/g, respectively [19].

Finally, Figure 5 shows the $N_2$ adsorption-desorption isotherms and pore size distribution of Ni,Pd,Pt/$Mg_{0.85}Ce^{4+}_{0.15}O$ catalysts. In Figure 5A, the isotherms at lower relative pressure ($\leq$0.3) have lower nitrogen adsorption (lines close to X axis), indicating a weak interaction between the nitrogen and catalyst, which can be ascribed to the type II, a characteristic of microporous materials. In Figure 5B, the narrow pore diameter distribution of these catalyst exhibited no peak.

### 2.1.5. Temperature Programmed Reduction ($H_2$-TPR)

The cerium reducibility for the Ni catalysts reforming was characterized using $H_2$-TPR. The $H_2$-TPR profiles for Ni/$Mg_{1-x}Ce^{+4}_xO$ (x = 0.00, 0.03, 0.07, 0.15) are illustrated in Table 3 along with the patterns of $H_2$-TPR for the catalysts in Figure 6A(a). First, using the $H_2$-TPR for the $CeO_2$ promoter, two peaks were recorded at 459 °C and 871 °C with $H_2$- consumption of 72.25 μmol/g. The peak observed at the temperature of 475 °C by the catalyst Ni/MgO was attributed to the Ni–O crystallite reduction [20]. Figure 6A(a) demonstrates the $H_2$-TPR profiles for Ni/$Mg_{0.97}Ce^{4+}_{0.03}O$, Ni/$Mg_{0.93}Ce^{4+}_{0.07}O$, and Ni/$Mg_{0.85}Ce^{4+}_{0.15}O$. Firstly, the peak formed for Ni/$Mg_{0.97}Ce^{4+}_{0.03}O$, Ni/$Mg_{0.93}Ce^{4+}_{0.07}O$, and Ni/$Mg_{0.85}Ce^{4+}_{0.15}O$ at temperatures of 279 °C, 283 °C, and 310 °C, respectively, was ascribed to the depletion of Ni–O to $Ni^0$. Secondly, the peak for Ni/$Mg_{0.97}Ce^{4+}_{0.03}O$, Ni/$Mg_{0.93}Ce^{4+}_{0.07}O$, and Ni/$Mg_{0.85}Ce^{4+}_{0.15}O$ was observed at temperatures of 480 °C, 488 °C, and 529 °C respectively, corresponding to the $CeO_2$ reduction on the surface of the Ni/$Mg_{1-x}Ce^{4+}_xO$ catalysts. Meanwhile, the third peak was formed at temperatures 695 °C, 701 °C, and 710 °C for Ni/$Mg_{0.97}Ce^{4+}_{0.03}O$, Ni/$Mg_{0.93}Ce^{4+}_{0.07}O$, and Ni/$Mg_{0.85}Ce^{4+}_{0.15}O$, respectively, and was attributed to the reduction of $CeO_2$ in the bulk of the catalysts. The lowering in the temperature of the second peak as compared to the third peak was due to the lowering in the reduction enthalpies. The first possible reason for this may be a result of the integration of MgO into $CeO_2$ and the hindrance of sintering that may have enhanced the dispersion of $CeO_2$ particles [21]. The other possible explanation may be due to the strong $CeO_2$ and Ni interaction [10]. It was concluded that the $H_2$-consumption of 314.8 μmol/g catalyst was implemented to reduce the total Ni-O to Ni on Ni/MgO. The total $H_2$-consumption amount of the reduced Ni/$Mg_{0.97}Ce^{4+}_{0.03}O$, Ni/$Mg_{0.93}Ce^{4+}_{0.07}O$, and Ni/$Mg_{0.85}Ce^{4+}_{0.15}O$ catalysts was observed at 415.0, 623.7, and 706.5 μmol/g, respectively, as calculated from the three peaks area indicating a

possible Ni–O reduction, and a partial $CeO_2$ reduction. An enhancement in the reducibility of the catalysts was observed following the addition of the $CeO_2$ promoter. It was observed that by elevating the concentration of the Ce promoter in the catalyst, the peaks appeared at high temperature indicating a good interaction between the catalyst's constituents [22].

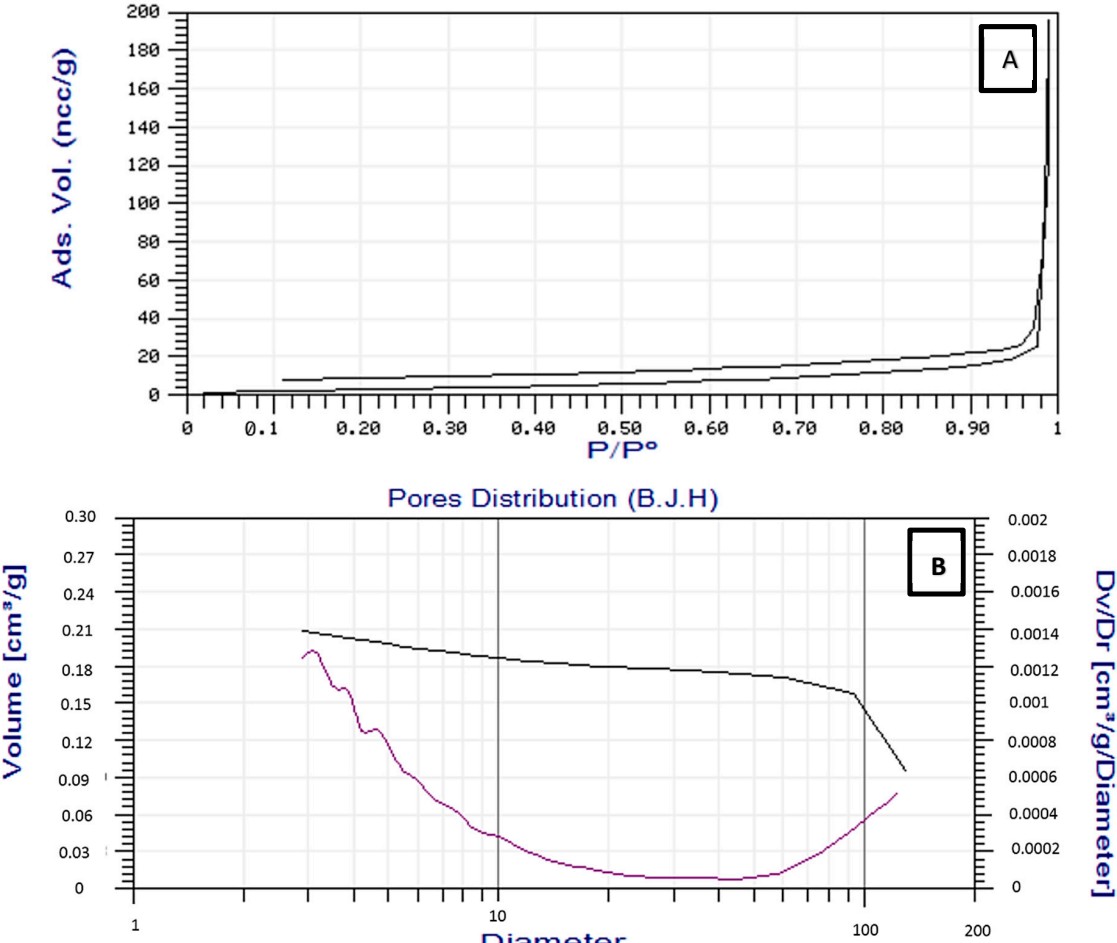

**Figure 5.** $N_2$ adsorption/desorption isotherms (**A**) and pore size distributions (**B**) of Ni,Pd,Pt/$Mg_{0.85}Ce^{4+}_{0.15}O$ catalyst.

**Table 3.** The different catalysts and their $H_2$-TPR values (reduced in a 5% $H_2$/Ar stream at a 10 °C/min temperature ramp).

| Catalysts | Temp. | Temp. | Temp. | Temp. | Temp. | Amount $H_2$-Consumed |
|---|---|---|---|---|---|---|
| | °C | °C | °C | °C | °C | (μmol/g) |
| $CeO_2$ | | | | 459 | 871 | 72.25 |
| Ni/MgO | | | 475 | | | 314.8 |
| Ni/$Mg_{0.97}Ce^{4+}_{0.03}O$ | | | 279 | 480 | 695 | 415.0 |
| Ni/$Mg_{0.93}Ce^{4+}_{0.07}O$ | | | 283 | 488 | 701 | 623.7 |
| Ni/$Mg_{0.85}Ce^{4+}_{0.15}O$ | | | 310 | 529 | 710 | 706.5 |
| Ni,Pd,Pt/MgO | 130 | 184 | 520 | | | 488.6 |
| Ni,Pd,Pt/$Mg_{0.97}Ce^{4+}_{0.03}O$ | 122 | 164 | 485 | 516 | 719 | 524.3 |
| Ni,Pd,Pt/$Mg_{0.93}Ce^{4+}_{0.07}O$ | 134 | 166 | 483 | 522 | 734 | 667.0 |
| Ni,Pd,Pt/$Mg_{0.85}Ce^{4+}_{0.15}O$ | 139 | 178 | 475 | 527 | 783 | 782.3 |

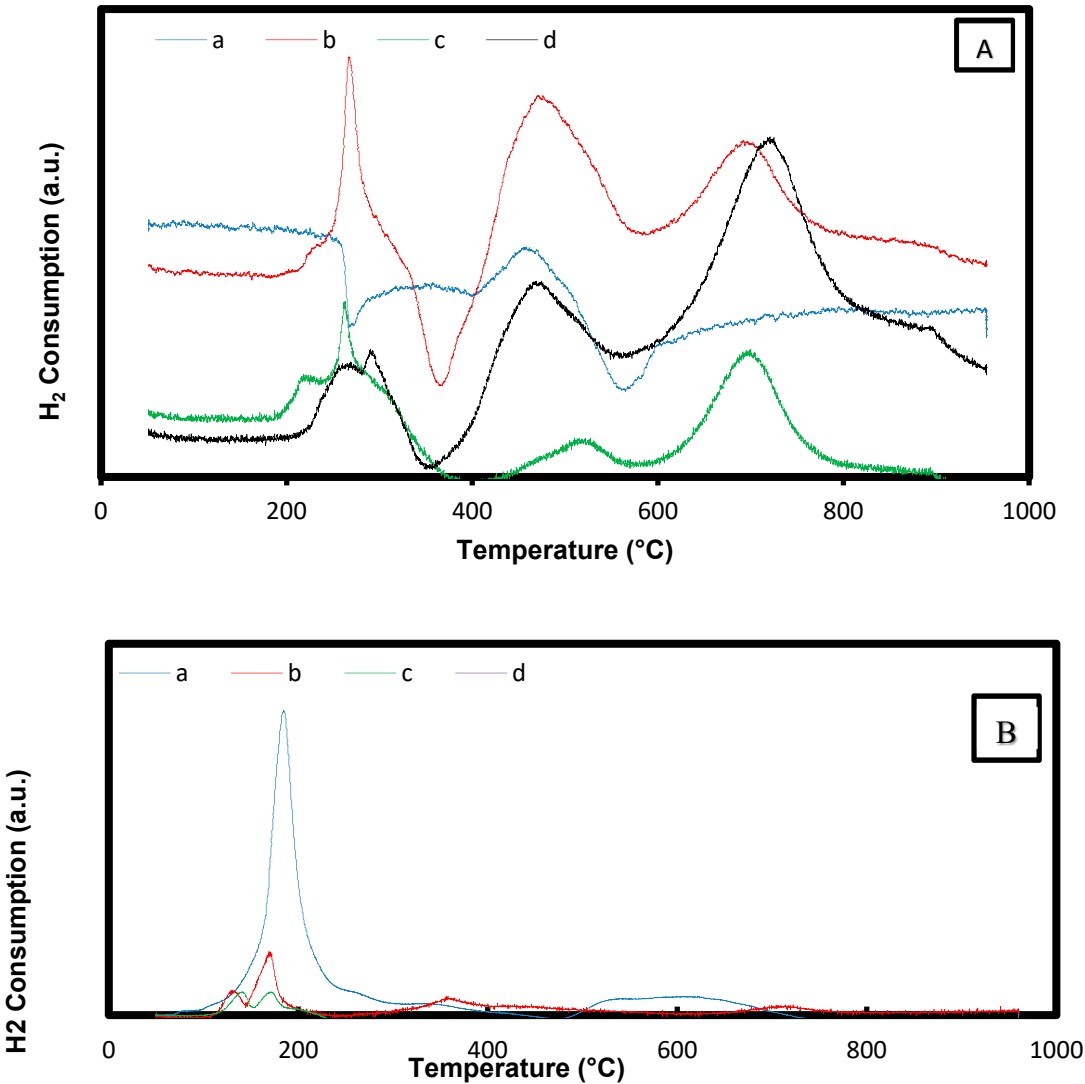

**Figure 6.** $H_2$-TPR profiles of (**A**) Ni/Mg$_{1-x}$Ce$^{4+}_x$O and (**B**) Ni,Pd,Pt/Mg$_{1-x}$Ce$^{4+}_x$O catalysts, were (x = a (0.00), b (0.03), c (0.07), and d (0.15)) (reduced in a 5% $H_2$/Ar stream at a 10 °C/min temperature ramp).

The profiles of $H_2$-TPR for the catalysts Ni,Pd,Pt/MgO, Ni,Pd,Pt/Mg$_{0.97}$Ce$^{4+}_{0.03}$O, Ni,Pd,Pt/Mg$_{0.93}$Ce$^{4+}_{0.07}$O, and Ni,Pd,Pt/Mg$_{0.85}$Ce$^{4+}_{0.15}$O are demonstrated in Table 3 and Figure 6B(a–d). The $H_2$-TPR patterns of these catalysts slightly differed from the previous catalysts. Figure 6B(a) showed three reduction peaks that were well-defined in the $H_2$-TPR profile of Ni,Pd,Pt/MgO. Th peak observed at 130 °C was a result of the PtO species reduction during the production of Pt$^0$. This finding contradicted with results by Mahoney et al., who detected the peak at 114 °C [23]. The second reduction peak was centered at 184 °C and was attributed to the PdO reduction to Pd$^0$. The final peak was observed at 520 °C and was linked to the reduction in the NiO species that resulted in a strong interaction with the supporting material to produce Ni. The NiO reduction temperature in the Ni/CeMgAl catalyst was reported to be 516 °C [24].

From Figure 6B(b–d) and Table 3, the TPR profiles of the Ni,Pd,Pt/Mg$_{0.97}$Ce$^{4+}_{0.03}$O, Ni,Pd,Pt/Mg$_{0.93}$Ce$^{4+}_{0.07}$O, and Ni,Pd,Pt/Mg$_{0.85}$Ce$^{4+}_{0.15}$O catalysts displayed five peaks that differed slightly from the Ni,Pd,Pt/MgO catalyst. The first three peaks of the Ni,Pd,Pt/Mg$_{0.97}$Ce$^{4+}_{0.03}$O catalyst were recorded at 122 °C, 164 °C, and 485 °C, whereas, the peaks for the Ni,Pd,Pt/Mg$_{0.93}$Ce$^{4+}_{0.07}$O catalyst were recorded at 134 °C, 166 °C, and 483 °C. The peaks of the Ni,Pd,Pt/Mg$_{0.85}$Ce$^{4+}_{0.15}$O catalyst were recorded at 139 °C, 178 °C, and 475 °C due to the NiO, PdO, and PtO reduction on the

catalysts surface to obtain the elements $Ni^0$, $Pd^{0,}$ and $Pt^0$ respectively. The $Ni,Pd,Pt/Mg_{0.97}Ce^{4+}_{0.03}O$, $Ni,Pd,Pt/Mg_{0.93}Ce^{4+}_{0.07}O$, and $Ni,Pd,Pt/Mg_{0.85}Ce^{4+}_{0.15}O$ catalysts fourth peaks were recorded at 516 °C, 522 °C, and 527 °C, respectively, corresponding to the $CeO_{2'}$ reduction on the catalyst's surface. A significant reduction in the surface of $CeO_2$ at a lower temperature of the catalysts was observed possibly due to the better $CeO_2$ particles dispersion during the integration of MgO into $CeO_2$ and the hindrance of sintering [10]. The fifth peak was recorded at temperatures of 719 °C, 734 °C, and 783 °C and was due to the bulk $CeO_2$ reduction that resulted in strong interactions between the support MgO and the $CeO_2$ promoter species. Enhanced reducibility along with an elevated promoter loading was observed by the catalysts. This result concurred with the findings on cerium reduction by Rotaru et al. [25] in which the cerium reduction occurred at 490 °C and 790 °C. A good promoter dispersion to the support induced a high level of interaction between the support and the doping Ni, Pd, and Pt species. The significant $H_2$-TPR profile peak recorded at temperatures of 684–737 °C proved that the temperature ranges can be lowered by $CeO_2$ alone [22]. It was also noted that adding the $CeO_{2,}$ promoter was efficient in the reducibility of the MgO-supported catalysts, which may be linked to the support's acidic-basic properties. It has been observed that $Mg_{1-x}Ce^{4+}_xO$ (higher basicity than MgO) interacted with the $CeO_2$ promoter. Hence, the reductions in NiO, PdO, and PtO, were clearer due to the redox property of $Mg_{1-x}Ce^{4+}_xO$ [26].

The total peaks area was used to measure the total $H_2$-consumption amount in the reduction of catalysts in which the amounts of the catalysts $Ni,Pd,Pt/MgO$, $Ni,Pd,Pt/Mg_{0.97}Ce^{4+}_{0.03}O$, $Ni,Pd,Pt/Mg_{0.93}Ce^{4+}_{0.07}O$, and $Ni,Pd,Pt/Mg_{0.85}Ce^{4+}_{0.15}O$ were 488.6, 524.3, 667.0, and 782.3 μmol/g, respectively. According to the $H_2$-consumption results of the $H_2$-TPR, the $H_2$ consumption with the highest value indicated a higher activity of the $Ni,Pd,Pt/Mg_{0.85}Ce^{4+}_{0.15}O$ catalyst when compared to other catalysts. Hence, the $Ni,Pd,Pt/Mg_{0.85}Ce^{4+}_{0.15}O$ catalyst was reported to have the most active site. In other words, the $Ni,Pd,Pt/Mg_{0.85}Ce^{4+}_{0.15}O$ catalyst was the best catalyst for the CH4 dry reforming.

### 2.1.6. Thermal Analysis

The catalysts $Ni/MgO$, $Ni/Mg_{0.97}Ce^{4+}_{0.03}O$, $Ni/Mg_{0.93}Ce^{4+}_{0.07}O$, and $Ni/Mg_{0.85}Ce^{4+}_{0.15}O$ components are illustrated by Figure 7A(a–d). At first, due to the $N_2$ gas adsorption on the compound, a slight increase was noted in weight. A weight loss of 1.8%, 2.0%, 2.2%, and 2.8%, was recorded for the $Ni/MgO$, $Ni/Mg_{0.97}Ce^{4+}_{0.03}O$, $Ni/Mg_{0.93}Ce^{4+}_{0.07}O$, and $Ni/Mg_{0.85}Ce^{4+}_{0.15}O$ catalysts, respectively, which may have been a result of the moisture removal from the $Ni/Mg_{1-x}Ce_xO$ catalysts. Thermal stability for all the catalysts was achieved at 500 °C, establishing a good interaction among the components of the catalyst [10]. On the other hand, Figure 7B(a-d) demonstrates the analysis of TG for the reduced catalysts: $Ni,Pd,Pt/MgO$, $Ni,Pd,Pt/Mg_{0.97}Ce^{4+}_{0.03}O$, $Ni,Pd,Pt/Mg_{0.93}Ce^{4+}_{0.07}O$, and $Ni,Pd,Pt/Mg_{0.85}Ce^{4+}_{0.15}O$. The results indicated that all the catalysts lost weight at only one stage referring to the loss of oxygen atom.

The estimated weight loss was approximately 2–4%, which was observed at temperatures between 100 °C and 150 °C due to the loss of humidity as displayed in Figure 7B(a–d). In contradiction, the estimated weight loss of 1.5%, 2.2%, 2.8%m and 4.0% was recorded for the $Ni, Pd,Pt/MgO$, $Ni,Pd,Pt/Mg_{0.97}Ce^{4+}_{0.03}O$, $Ni,Pd,Pt/Mg_{0.93}Ce^{4+}_{0.07}O$, and $Ni,Pd,Pt/Mg_{0.85}Ce^{4+}_{0.15}O$ catalysts, respectively.

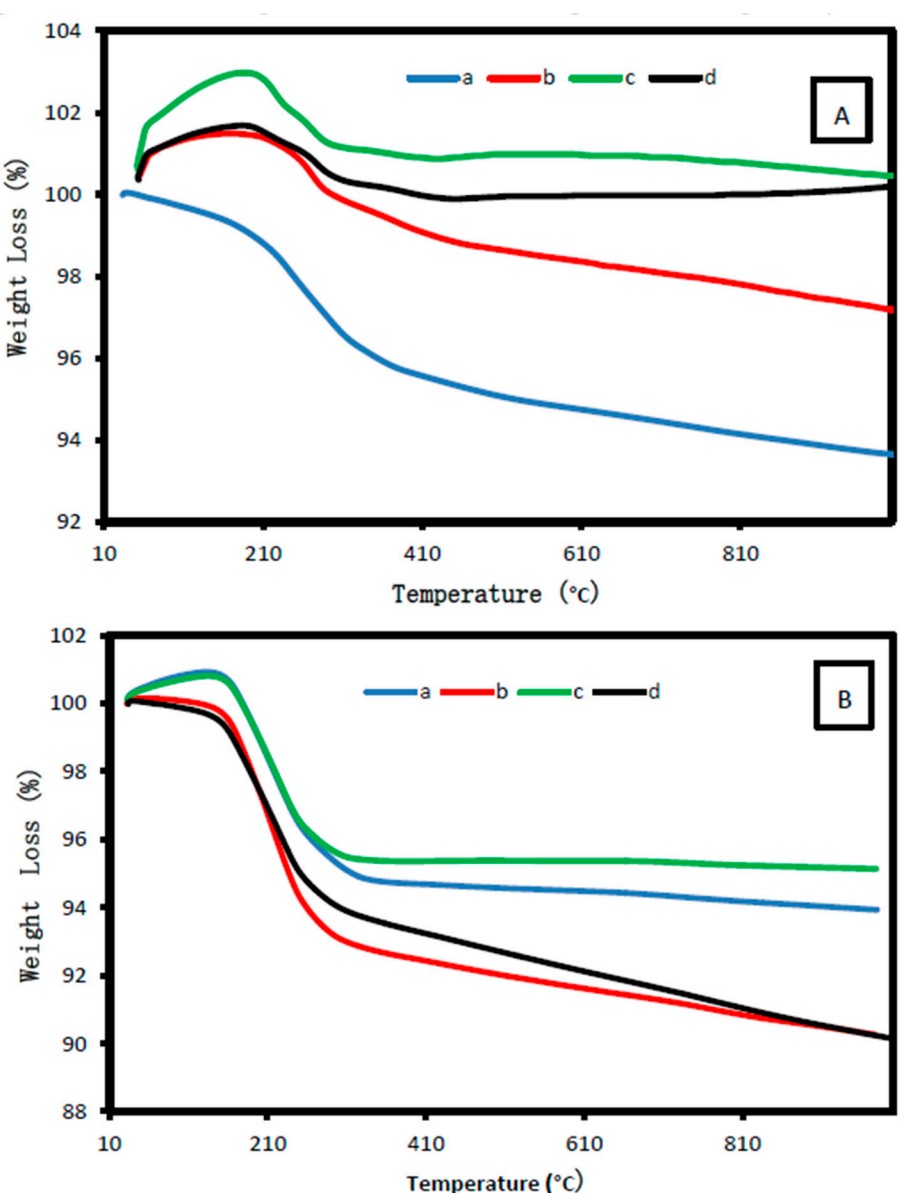

**Figure 7.** TGA of (**A**) Ni/Mg$_{1-x}$Ce$^{4+}$$_x$O and (**B**) Ni,Pd,Pt/Mg$_{1-x}$Ce$^{4+}$$_x$O catalysts, where (x = a (0.00), b (0.03), c (0.07), and d (0.15)).

## 2.2. The Performance of Catalysts in DRM Reaction

### 2.2.1. Effects of the Concentration of Reactants on the Conversion

The CH$_4$ and CO$_2$ conversion, and the selectivity (H$_2$/CO ratio) revealed the activity of the dry reforming reaction. Upon elevating the temperature to 900 °C, blank tests (reaction without catalyst) displayed the existence of H$_2$ and CO in the outlet gas, which may be attributed to the reaction of methane decomposition reaction (Equation (3)). Using Mg$_{1-x}$Ce$_x$O without the main catalyst (metals) resulted in lowering the conversion of CH$_4$ (32%) and CO$_2$ (41%) with an H$_2$/CO ratio of 0.3% indicating a possibility of the weak reaction on the promoter-support pores as presented by the BET results. Contrarily, when using the catalyst Ni,Pd,Pt/Mg$_{1-x}$Ce$^{4+}$$_x$O, or Ni/Mg$_{1-x}$Ce$^{4+}$$_x$O, an enhancement in the rate of CH$_4$ and CO$_2$ conversion and the H$_2$/CO ratio was noted. The effect of the reactant ratio (CH$_4$: CO$_2$) on the CH$_4$, CO$_2$ conversion, and H$_2$/CO ratio is illustrated in Figure 8. In the reaction, two ratios of (CH$_4$: CO$_2$) were used, 1:1 and 2:1. By increasing the CO$_2$ concentration in the (CH$_4$: CO$_2$) ratio to 1:1, the CO$_2$ and CH$_4$ conversion and the H$_2$/CO ratio were increased. This may have attributed

to the decline in the deposition of carbon on the catalyst that reacted with the excess $CO_2$ to yield CO (Equation (5)). Furthermore, the doped Ni, Pd, and Pt metals on the promoter-support played an imperative function in the catalytic reaction. It has been observed that the most $CH_4$ (78%) and $CO_2$ (90%) conversion was by Ni,Pd,Pt/Mg$_{0.85}$Ce$^{4+}_{0.15}$O catalyst with a $CH_4$: $CO_2$ (1:1), and an $H_2/CO$ ratio of 1.1. However, at a ratio of 2:1, the conversion of the gases $CH_4$ and $CO_2$ was recorded at 70% and 82%, respectively, with a 0.8 $H_2/CO$ ratio. This finding demonstrated that the best deactivate resistance of the catalyst stands at a 1:1 ratio due to the decline in carbon formation, which leads to a high $H_2$ and CO selectivity (Figure 7). Similar results were also acquired by the other reported catalysts [27].

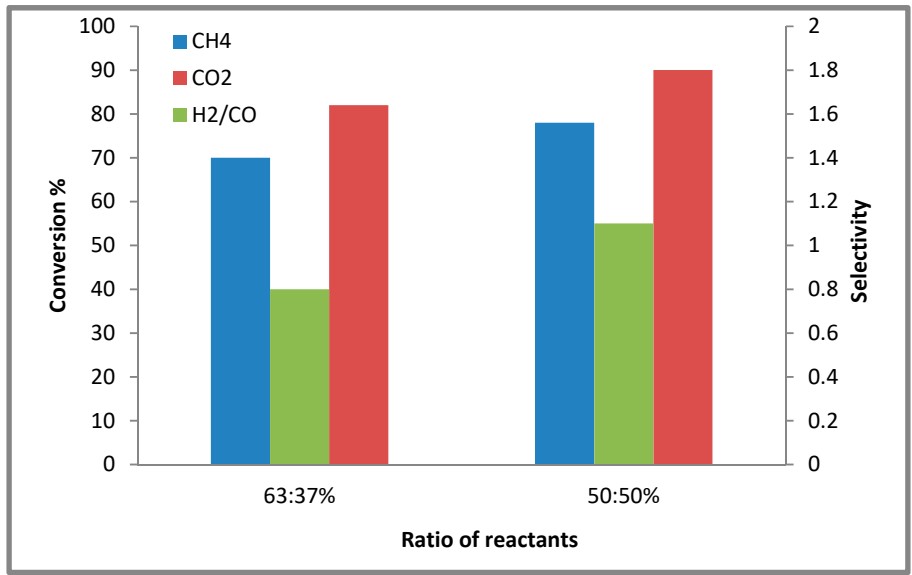

**Figure 8.** Changing the reactant $CH_4$:$CO_2$ ratio concentration 1–2:1 2–1:1 over the conversion percentage and $H_2/CO$ ratio for Ni,Pd,Pt/Mg$_{0.85}$Ce$^{4+}_{0.15}$O catalyst at 900 °C, with GHSV=15,000 mL h$^{-1}$g$^{-1}$cat.

### 2.2.2. Effect of the Catalyst Concentration on the Conversion

The role of the catalyst concentrations on the process of conversion is illustrated by Figure 9 and Table 4. The $CH_4$ and $CO_2$ conversion values and $H_2/CO$ ratio for the catalyst Ni,Pd,Pt/MgO were 72%, 81%, and 0.7, whereas for the catalyst Ni,Pd,Pt/Mg$_{0.97}$Ce$^{4+}_{0.03}$O, the values were 73%, 86%, and 0.8 and the values for the Ni,Pd,Pt/Mg$_{0.93}$Ce$^{4+}_{0.07}$O catalyst were 76%, 88%, and 0.9. The highest values for catalyst Ni,Pd,Pt/Mg$_{0.85}$Ce$^{4+}_{0.15}$O were 78%, 90%, and 1.1 for the $CH_4$ and $CO_2$ conversion and $H_2/CO$ ratio.

**Table 4.** The DRM reaction results of the catalysts at 900 °C and a $CH_4$:$CO_2$ ratio of 1:1 ratio with GHSV = 15,000 mL h$^{-1}$g$^{-1}$cat.

| Sample Name | $CH_4$ Conversion % | $CO_2$ Conversion % | $H_2/CO$ Ratio |
|---|---|---|---|
| MgO | 21 | 32 | 0.1 |
| Mg$_{0.85}$Ce$^{4+}_{0.15}$O | 32 | 41 | 0.3 |
| Ni/MgO | 45 | 66 | 0.4 |
| Ni/Mg$_{0.97}$Ce$^{4+}_{0.03}$O | 40 | 67 | 0.5 |
| Ni/Mg$_{0.93}$Ce$^{4+}_{0.07}$O | 48 | 66 | 0.5 |
| Ni/Mg$_{0.85}$Ce$^{4+}_{0.15}$O | 56 | 74 | 0.6 |
| Ni,Pd,Pt/MgO | 72 | 81 | 0.7 |
| Ni,Pd,Pt/Mg$_{0.97}$Ce$^{4+}_{0.03}$O | 73 | 86 | 0.8 |
| Ni,Pd,Pt/Mg$_{0.93}$Ce$^{4+}_{0.07}$O | 76 | 88 | 0.9 |
| Ni,Pd,Pt/Mg$_{0.85}$Ce$^{4+}_{0.15}$O | 78 | 90 | 1.1 |

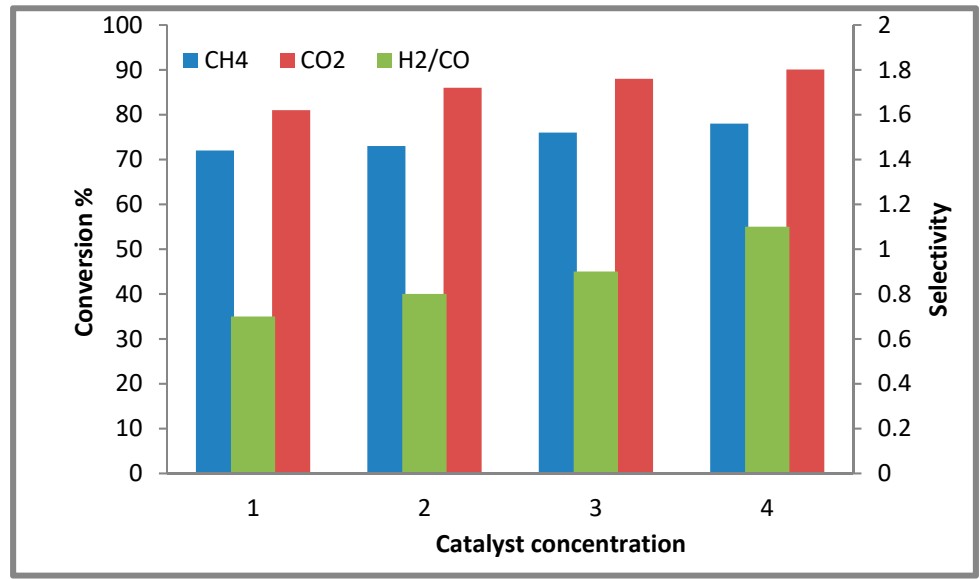

**Figure 9.** The effect of utilizing (**1**) Ni,Pd,Pt/MgO, (**2**) Ni,Pd,Pt/Mg$_{0.97}$Ce$^{4+}_{0.03}$O (**3**) Ni,Pd,Pt/Mg$_{0.93}$Ce$^{4+}_{0.07}$O, and (**4**) Ni,Pd,Pt/Mg$_{0.85}$Ce$^{4+}_{0.15}$O catalysts on CH$_4$, CO$_2$ conversion, and H$_2$/CO ratio at CH$_4$:CO$_2$ ratio of 1:1 ratio and a temperature of 900 °C, with GHSV = 15,000 mL h$^{-1}$g$^{-1}$cat.

In this study, the positive results may have been a result of the favorable interaction between Pt, Pd, and Ni metals and promoter-support and the good basicity of the promoter-support. Unfavorable results were previously reported by Laosiripojana [28] and Guo et al. [29] due to the usage of Ni/Al$_2$O$_3$ catalyst, which demonstrated a weak Ni and support interaction and low basicity of Al$_2$O$_3$.

The decreasing order of CH$_4$ and CO$_2$ conversion and the H$_2$/CO ratio can be described as Ni,Pd,Pt/Mg$_{0.85}$Ce$^{4+}_{0.15}$O > Ni,Pd,Pt/Mg$_{0.93}$Ce$^{4+}_{0.07}$O > Ni,Pd,Pt/Mg$_{0.97}$Ce$^{4+}_{0.03}$O > Ni,Pd,Pt/MgO. This indicates that the most efficient catalyst was Ni,Pd,Pt/Mg$_{0.85}$Ce$_{0.15}$O. The results illustrated that the formation rate of H$_2$ and CO gases in the DRM reaction relies on the amount of solid solution MgO-CeO$_2$ in the catalyst. As such, the larger the amount of MgO-CeO$_2$ solid solution, the greater the rate of formation of H$_2$ and CO gases is. Hence, the formation of the MgO-CeO$_2$ solid solution is critical in the active site generation for the DRM reaction. This happens because the entire CeO$_2$ is like a solid solution, which stabilizes both oxides. Only the surface layer of CeO$_2$ of the catalyst of the solid solution, MgO-CeO$_2$, was reduced at 700 °C. The sites of Ce that were formed remained close to the solid solution, hindering the sintering of Ce [25]. Table 4 illustrates the activity and selectivity of Ni,Pd,Pt/Mg$_{1-x}$Ce$^{4+}_x$O, which was more than that of Ni/Mg$_{1-x}$Ce$^{4+}_x$O in the DRM reaction. This might be due to the presence of platinum and palladium metals that provides more electrons to Nickel resulting in an elevation in the electron density of nickel and hence inhibiting the catalyst coke sintering and preventing nickel oxidation.

Moreover, the enhancement in the CH$_4$ and CO$_2$ conversion rate was ascribed to the particle size involved in the activity of the reaction. Using the equation of Debye Sherrer's and TEM analysis, the Ni, Pd, and Pt doping metals were prepared with a particle size as minuscule as nanoparticles. Hence, the crucial role of the particle size is evident in the reaction activity. An enhancement in the reactants conversion and selectivity (yield) may be due to the reduction of the particles into nano-ranged sizes, along with having the highest BET surface area (19.8 m$^2$/g) (Table 2) and the highest H$_2$-consumption in H$_2$-TPR (782.3 µmol/g of active sites) (Table 3).

### 2.2.3. The Effect of Various Temperatures on the Conversion

The activity and selectivity of the Ni,Pd,Pt/Mg$_{0.85}$Ce$^{4+}_{0.15}$O catalyst at temperatures from 700–900 °C can be seen in Figure 10. Generally, an enhancement in the CH$_4$: CO$_2$ conversion of (1:1) was noted upon increasing the temperature from 700 °C to 900 °C, which may be attributed

to the strong endothermic reaction of dry-reforming (Equation (1)). Earlier research reported an increase in the rate of conversion at higher temperatures [30]. It is demonstrated that an increase in the temperature (700–900 °C) led to an increase in the $CH_4$ conversion of $Ni,Pd,Pt/Mg_{0.85}Ce^{4+}_{0.15}O$ (33% to 78%) and an increase in the $CO_2$ conversion from 41% to 90%. However, at a temperature of more than 900 °C, no evident elevation in the $CH_4$ and $CO_2$ conversion rates was observed. Figure 9 illustrates the catalyst $H_2/CO$ ratio at a range of temperatures. At a temperature lower than 900 °C, a <1 $H_2/CO$ ratio of the samples was observed. The reverse water-gas-shift reaction (RWGS), (Equation (2)) might consume the extra $H_2$ to produce CO, leading to a mitigation in the $H_2/CO$ ratio. At a 900 °C temperature, the $H_2/CO$ ratio of the $Ni,Pd,Pt/Mg_{0.85}Ce^{4+}_{0.15}O$ catalyst was recorded at 1.1, demonstrating a slight contribution from the reaction of RWGS (Equation (2)) [31].

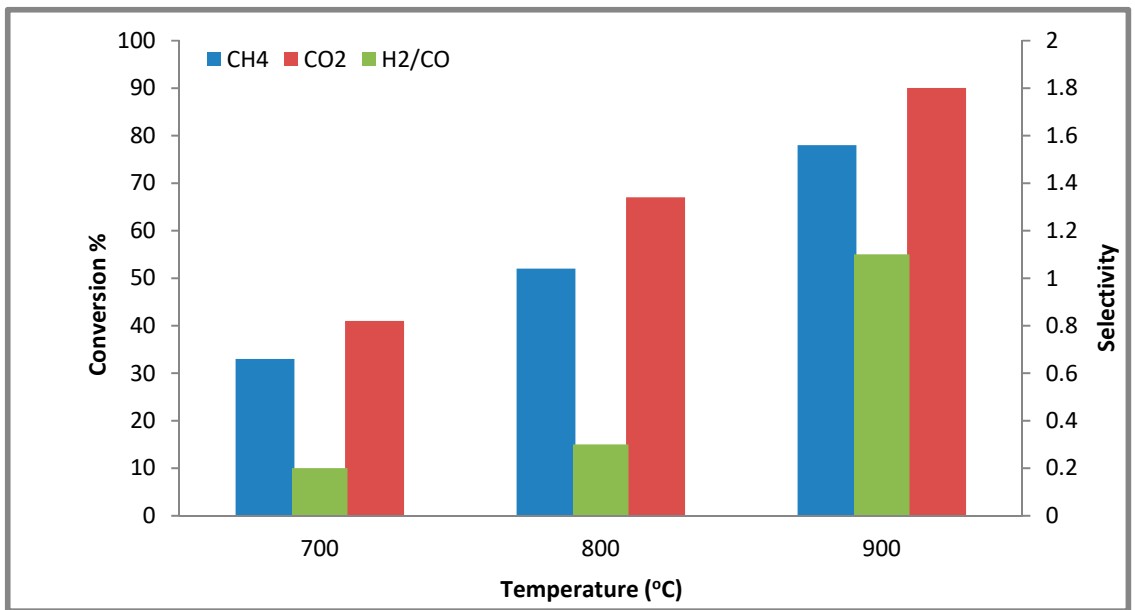

**Figure 10.** The effect of various temperatures on the activity of $Ni,Pd,Pt/Mg_{0.85}Ce^{4+}_{0.15}O$ catalyst. 700 °C; 800 °C; 900 °C at a $CH_4:CO_2$ ratio of 1:1, with GHSV = 15,000 mL $h^{-1}g^{-1}$cat.

### 2.2.4. Stability Tests

Figure 11 At a 900 °C temperature, the reaction between $CO_2$ and $CH_4$ was achieved for a long duration. At first, Methane adsorbed on the surface of nickel surface of the catalyst to yield hydrogen resulting in the carbon deposition on the surface of nickel as can be seen (Equations (6)–(10)) [32].

$$CH_4 + 2Ni_{as} \rightarrow CH_3Ni_{as} + HNi_{as} \tag{6}$$

$$CH_3Ni_{as} + Ni_{as} \rightarrow CH_2Ni_{as} + HNi_{as} \tag{7}$$

$$CH_2Ni_{as} + Ni_{as} \rightarrow CHNi_{as} + HNi_{as} \tag{8}$$

$$CHNi_{as} + Ni_{as} \rightarrow CNi_{as} + HNi_{as} \tag{9}$$

$$2HNi_{as} \rightarrow H_2 + 2Ni_{as} \tag{10}$$

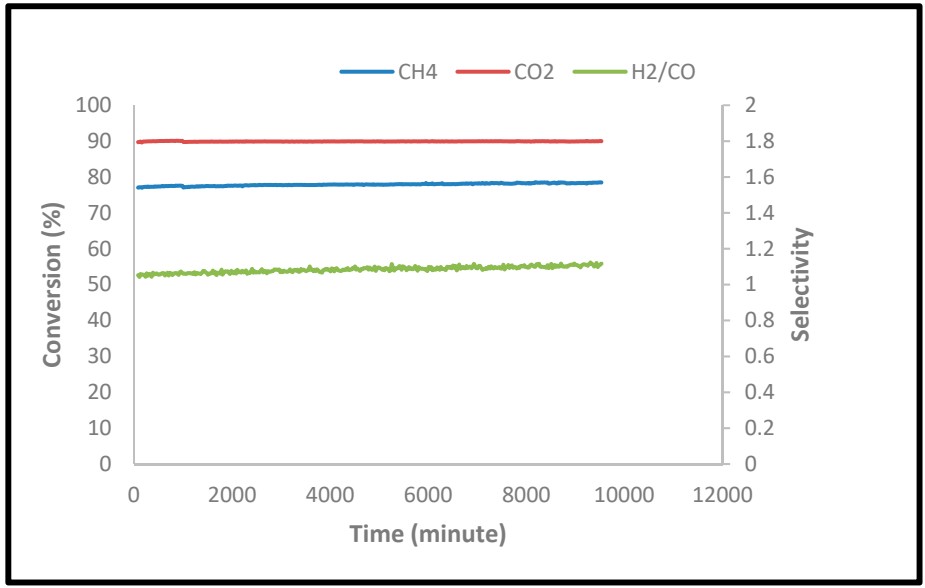

**Figure 11.** Ni,Pd,Pt/Mg$_{0.85}$Ce$^{4+}_{0.15}$O catalysts stability tests with a temperature of 900 °C, GHSV = 15,000 mL h$^{-1}$g$^{-1}$cat., and CH$_4$/CO$_2$ ratio of 1:1. atmospheric pressure, for 200 h.

Secondly, Topalidis et al. demonstrated the effect of the catalyst promoter (CeO$_2$) on dry reforming of methane [32]. In line with the mechanisms, the activation of CO$_2$ took place on the Ce metal particle (Equations (11)–(15)):

$$CO_{2_g} \rightarrow CO_{2_{support}} \tag{11}$$

$$CO_{2_{g_{support}}} + O^{-2}_{support} \rightarrow CO_{3_{support}}{}^{-2} \tag{12}$$

$$2H_{metal} \rightarrow 2H_{support} \tag{13}$$

$$CO_{3_{support}}{}^{-2} + 2H_{support} \rightarrow HCO_{2_{support}}{}^{-1} + OH^{-1}_{support} \tag{14}$$

$$CO_{support} \rightarrow CO_g \tag{15}$$

The buildup of carbon on the nickel metal surface has been known to curb the catalytic stability that is counteracted by the CeO$_2$ promoter availability. This facilitates clearing out the deposited carbon resulting in catalytic reactivation. The main reason for the continuation of the reaction for a long time (>200 h) was utilizing the CeO$_2$ promoter in the catalyst. CeO$_2$ ensured strong coke resistance and a very stable platform. CeO$_2$ also removed the carbon formed on the catalyst during the reaction of DRM. Subsequently, carbonate types (CeOCO$_3$) were formed especially CeO$_2$ (able to convert carbon dioxide into CO and O). Finally, an atom of oxygen was generated with C and was deposited on the Ni metal catalyst to yield CO. According to the results, the reduction in the catalytic carbon deposition was significantly noted (Equations (16) and (17)):

$$CO_{2_g} \rightarrow CO_{support} + O_{promoter} \tag{16}$$

$$C_{metal} + O_{promoter} \rightarrow CO_g \tag{17}$$

In conclusion, the above mechanism led to inhibiting the deposition of carbon on the Ni,Pd,Pt/Mg$_{0.85}$Ce$^{4+}_{0.15}$O catalysts surface making the catalyst fit for long-term usage.

Figure 12 illustrates the H$_2$ selectivity of the DRM reaction using Ni,Pd,Pt/Mg$_{0.85}$Ce$^{4+}_{0.15}$O catalysts.

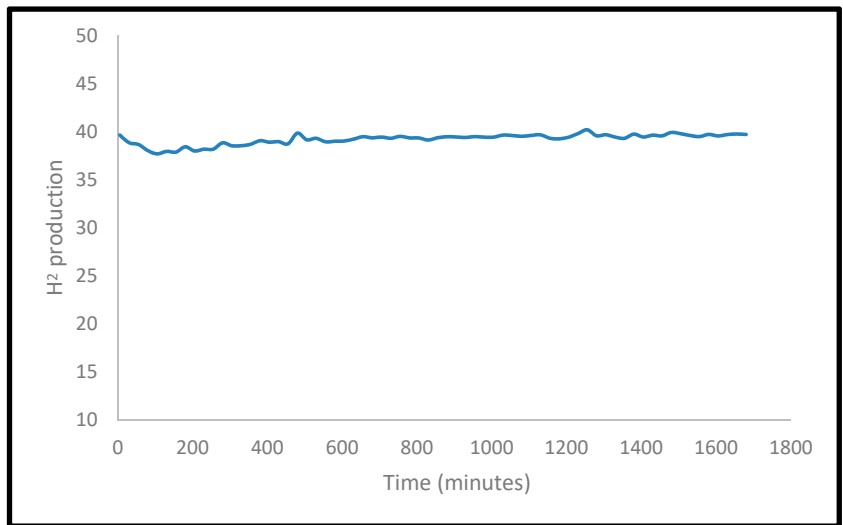

**Figure 12.** $H_2$ Selectivity of Ni,Pd,Pt/Mg$_{0.85}$Ce$^{4+}_{0.15}$O catalysts with a temperature of 900 °C, GHSV = 15,000 mL h$^{-1}$g$^{-1}$cat., and CH$_4$/CO$_2$ ratio of 1:1. atmospheric pressure.

### 2.2.5. Post-Reaction Characterization

The evaluation of coke formation on the catalyst Ni,Pd,Pt/Mg$_{0.85}$Ce$^{4+}_{0.15}$O took place by the TPO-MS, TEM images, and BET post-reaction tests. No coke deposition was observed on the catalyst surface as demonstrated by the TPO-MS profile (Figure 13). The analysis of TEM for the spent Ni,Pd,Pt/Mg$_{0.85}$Ce$^{4+}_{0.15}$O catalyst agreed with the finding in Figure 14. The original catalyst structure was maintained even following stream testing of 200 h as can be seen in the image. Moreover, the 2-D cubic texture of the spent catalyst was maintained. A slight increase in the spent catalyst pore size from 86.3 Å to 89.1 Å was reported. The BET analysis also demonstrated a marginal increase from 19.8 to 20.2m$^2$/g in the spent catalyst surface area. Due to the absence of filamentous carbon on the spent catalyst, it can be inferred that the coke deposition was insignificant. Zhu et al. reported that the smaller metal crystal size forms a catalyst less prone to deactivation [33]. The role of platinum and palladium metals for the spent catalyst was indicated by the TPO-MS and TEM results that showed no carbon deposition and no sintering of the spent catalyst.

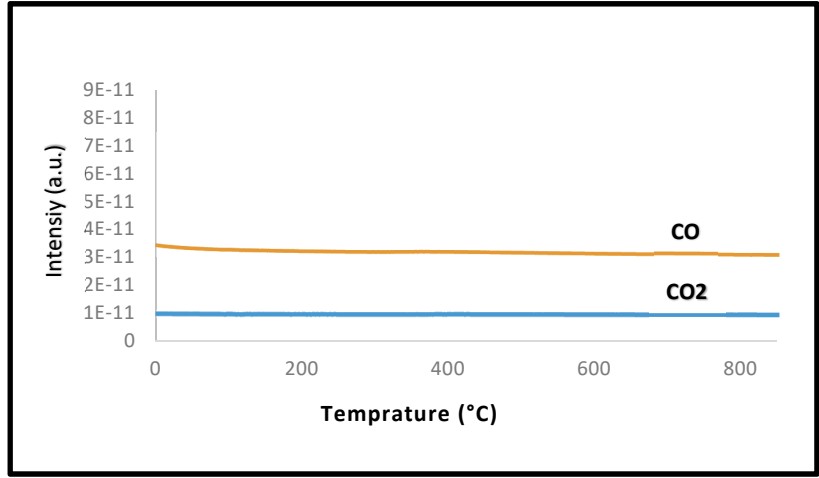

**Figure 13.** Temperature programmed oxidation TPO curves of spent Ni,Pd,Pt/Mg$_{0.85}$Ce$^{4+}_{0.15}$O catalyst after reaction at T = 900 °C with ratio of 1:1 CH$_4$/CO$_2$. and GHSV = 15,000 mL h$^{-1}$g$^{-1}$cat.

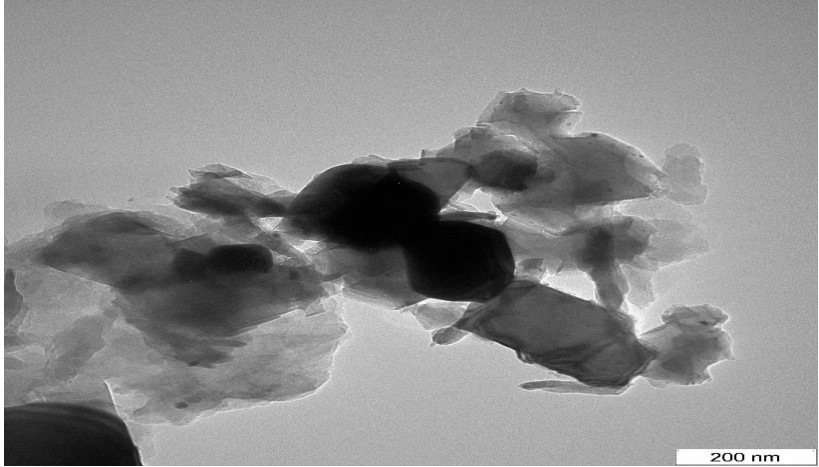

**Figure 14.** TEM analysis of the spent catalyst Ni,Pd,Pt/Mg$_{0.85}$Ce$^{4+}$$_{0.15}$O after reaction at T = 900 °C with ratio of 1:1. CH$_4$/CO$_2$. and GHSV = 15,000 mL h$^{-1}$g$^{-1}$cat.

### 2.2.6. Improvement in the Catalytic Stability and Selectivity

The reaction of DRM can be improved by using a low oxygen stream (1.25%). Figure 15 demonstrates an improvement in the CH$_4$ conversion (78% to 87%) following the addition of an oxidant (O$_2$) and utilizing the exothermicity of the reaction. The purpose of including an oxidant was to synthesize methane (partially or completely) whereas the use of exothermicity of the reaction was to provide the sufficient heat directly to the DRM reactant mixture [12]. No effect was observed on the CO$_2$ conversion and the H$_2$/CO ratio, which may be attributed to the reaction of CH$_4$ with oxygen to yield H$_2$O and CO (Equation (18)). Finally, syngas was produced following the reaction of steam with the deposited carbon (Equation (19)). Moreover, O$_2$ has the potential of oxidizing coke deposition on the catalyst (Equation (20)). Hence, the inclusion of this process has decreased the carbon deposition consequently improving the catalyst lifetime.

$$CH_4 + 3/2O_2 \rightarrow CO + 2H_2O \tag{18}$$

$$C + H_2O \rightarrow CO + H_2 \tag{19}$$

$$C + 1/2O_2 \rightarrow CO \tag{20}$$

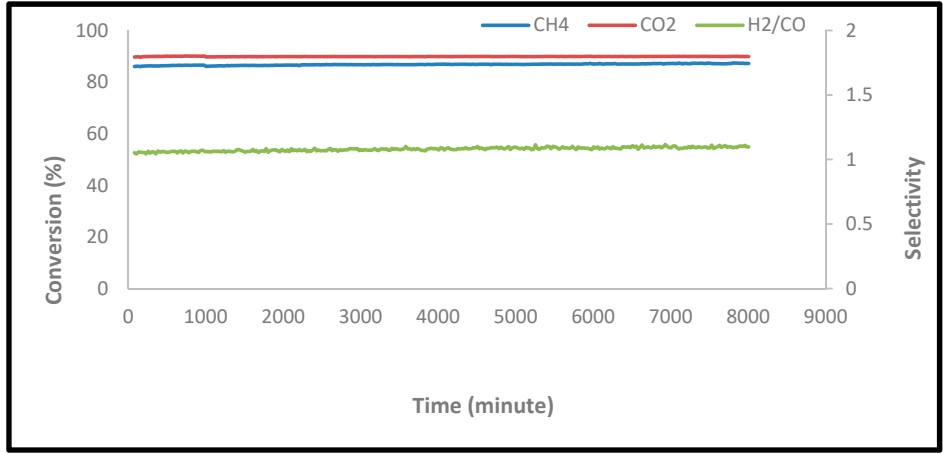

**Figure 15.** DRM reaction of the Ni,Pd,Pt/Mg$_{0.85}$Ce$^{4+}$$_{0.15}$O catalyst with a reaction conditions: T = 900 °C, CH$_4$/CO$_2$ ratio of 1:1, 1.25% O$_2$, and GHSV = 15,000 mL h$^{-1}$ g$^{-1}$cat.

## 3. Experimental Section

### 3.1. Materials

Ninety-nine percent $Mg(NO_3)_2.6H_2O$ and $(NH_4)_2Ce(NO_3)_6.6H_2O$ and 99.7% $K_2CO_3$ (were acquired from Merck (Merck, Kenilworth, NJ, USA). Ninety-nine percent $Ni(C_5H_7O_2)_2 \cdot H_2O$ and $Pt(C_5H_7O_2)_2 \cdot H_2O$ were purchased from Acros Organics (Acros Organics, Waltham, MA, USA), whereas 99.5% $Pd(C_5H_7O_2)_2 \cdot H_2O$ was acquired from Sigma-Aldrich (Sigma-Aldrich, St. Louis, MO, USA).

### 3.2. Catalysts Preparation

The co-precipitation method was implemented to prepare the promoter-supports $Mg_{1-x}Ce^{4+}_xO$ (x = 0.00, 0.03, 0.07, 0.15) [12]. Meanwhile, the MgO (support) and ($CeO_2$) promoter were developed using 0.1M solution of $Mg(NO_3)_2 \cdot 6H_2O$ and $(NH_4)_2Ce(NO_3)_6 \cdot 6H_2O$ and 1M $K_2CO_3$, which were used as precipitants. Following the filtration of the precipitant, soaking the sample in hot water took place. The sample was then dried for 12 h at 120 °C. Successively, pre-calcination of the sample took place in an open furnace at 500 °C for 5h to extract $CO_2$ from the precipitant. Samples were placed into disks at 600 kg/m$^2$, followed by calcination for 20 h at 1150 °C to increase the mechanical characteristics and ensure an efficient promoter and support interaction.

Primarily, impregnation of 1% Ni using $Ni(C_5H_7O_2)_2$ dissolved in dichloromethane took place. After air impregnation, the catalysts were dried (12 h at 120 °C temperature). Upon grinding, sieving the catalysts into particles of diameter 80–150 or 150–250 μm followed. Similar steps were applied for the $Ni(acac)_2/Mg_{1-x}Ce^{4+}_xO$ (x = 0.03, 0.07, 0.15) catalysts. Then, a small amount of the $Ni(acac)_2/Mg_{1-x}Ce^{4+}_xO$ catalysts was reduced by 30 ml min$^{-1}$ $H_2/Ar$ (5%) at 700 °C and 3 h holding. The remaining amount of the $Ni(acac)_2/Mg_{1-x}Ce^{4+}_xO$ catalyst was used to prepare $Ni,Pd,Pt/Mg_{1-x}Ce^{4+}_xO$.

Table 5 demonstrates the catalysts preparation ($Ni,Pd,Pt(acac)_2/Mg_{1-x}Ce_xO$). Initially, 1% Pt was impregnated on $Ni(acac)_2/Mg_{1-x}Ce^{4+}_xO$ using $Pt(C_5H_7O_2)_2$. $H_2O$ dissolved with dichloromethane for 5h to form $Pt(acac)_2,Ni(acac)_2/Mg_{1-x}Ce_xO$. Subsequently, the preparation of the $Ni,Pd,Pt(acac)_2/Mg_{1-x}Ce^{4+}_xO$ catalysts was achieved by impregnating $Pt(acac)_2$, $Ni(acac)_2/Mg_{1-x}Ce_xO$ with 1% Pd by using $Pd(C_5H_7O_2)_2$ solution in dichloromethane for 5 h. After air impregnation, at a temperature of 120 °C, the catalysts were dried for 12 h. Crushing and sieving the dried catalysts to particles of 80–150 or 150–250 μm diameter happened next. Finally, the reduction of the catalysts took place by flowing (5%) $H_2/Ar$ (30 mL min$^{-1}$) at 700 °C and holding for 3 h to yield the reduced $Ni,Pd,Pt/Mg_{1-x}Ce^{4+}_xO$ catalysts.

**Table 5.** Catalysts preparation with 0.1 M solutions of the mixed promoter and support followed by determining the total MgO weight with the promoter after precipitation with 1 M $K_2CO_3$ and calcination at 1150 °C.

| Catalysts | Support (MgO) Mg $(NO_3)_2.6H_2O$ (g) | Promoter ($CeO_2$) $(NH_4)_2Ce$ $(NO_3)_6.6H_2O$ (g) | Total Weight of MgO and $CeO_2$ after Calcination (g) | Impregnation of the Main Catalyst (1% Pt) (1% Pd) (1% Ni) (g) | | |
|---|---|---|---|---|---|---|
| | | | | Pt $(acac)_2.H_2O$ | Pd $(acac)_2$ | Ni $(acac)_2$ |
| Ni,Pd,Pt/MgO | 25.0 | 0.0 | 1 | 0.02 | 0.029 | 0.044 |
| Ni,Pd,Pt/Mg$_{0.97}$Ce$^{4+}_{0.03}$O | 24.9 | 1.3 | 1 | 0.02 | 0.029 | 0.044 |
| Ni,Pd,Pt/Mg$_{0.93}$ Ce$^{4+}_{0.07}$O | 23.8 | 3.0 | 1 | 0.02 | 0.029 | 0.044 |
| Ni,Pd,Pt/Mg$_{0.85}$ Ce$^{4+}_{0.15}$O | 21.8 | 6.5 | 1 | 0.02 | 0.029 | 0.044 |

### 3.3. Catalysts Characterization

The analysis of X-ray diffractometer, XRD, was performed using a Shimadzu diffractometer (Shimadzu XRD 6000, Chiyoda-ku, TYO, Japan). Debye–Scherrer's relationship was implemented to measure the samples size crystallite [34].

X-ray photoelectron spectroscopy (XPS) analysis was conducted using a Kratos Axis Ultra DLD system (Kratos Analytical Limited, Trafford Park, MCR, UK) with a $1 \times 10^{-10}$ Torr base pressure of the analyzer chamber.

The study of morphology by transmission electron microscopy (TEM) (Hitachi H7100, Chiyoda, TYO, Japan) accelerating voltage of 10 MV was implemented to study the shape of the crystal and the catalysts homogeneity.

The sample topology was determined using JSM 7600F Field Emission Scanning Electron Microscopy (FE-SEM) (JEOL Ltd., Akishima, TYO, Japan) using a field emission current at very high magnification. The sample was gold-coated to maintain the enhanced surface visibility and to exclude the sample electrical charging during analysis.

By applying the Brunauer–Emmett–Teller (BET) method with nitrogen gas adsorption ($-196$ °C), the total catalyst surface area was obtained. The analysis took place using a nitrogen adsorption-desorption analyzer (Surfer Analyzer) (Thermo Fisher Scientific, Rodano, Italy).

The catalyst's active site was evaluated using temperature-programmed reduction ($H_2$-TPR) Thermo Finnegan TPDRO 1100 (Thermo Fisher Scientific, Waltham, MA, USA) equipped with a detector of the thermal conductivity. Treatment of about 0.05 g of catalyst took place in the reactor at 150 °C for 30 min in $N_2$ (20 mL/min). The analysis of Hydrogen 5.51% in Argon was achieved at 50 °C and 950 °C under Argon flow (10 °C min$^{-1}$, 25 mL min$^{-1}$) and was detected by the detector of thermal conductivity.

Utilizing a Mettler Toledo TG-DTA Apparatus (Mettler-Toledo, Shah Alam, SGR, Malaysia) helped to conduct the thermogravimetric analysis (TGA), Pt crucibles, Pt/Pt-Rh thermocouple, with a purge gas (nitrogen) flow rate of 30 mL min$^{-1}$, and a heating rate of 10 °C/min from 50 to 1000 °C.

## 3.4. Catalytic Evaluations

A fixed-bed stainless-steel micro-reactor was used for the catalytic evaluation of DRM (i. d. $\varnothing$ = 6 mm, *h* = 34 cm). Preceding the reaction, the reduction of about 0.02 g catalyst took place by flowing $H_2$/Ar (5%) (30 mL min$^{-1}$) at 700 °C followed by holding for 3 h. The reforming reaction was performed by flowing the feed composed of $CH_4$: $CO_2$ in (2:1) and (1:1) mol, at a rate of 30 mL min$^{-1}$. The reforming was studied at 1 atm and a temperature range of 700–900 °C, followed by holding at 1 atm for 10 h (GHSV = 15,000 mL h$^{-1}$ g$^{-1}$cat).

Vertically, the studied catalyst was set in the middle of the reactor and fixed by quartz wool plugs. To regulate and ensure the temperature of the reaction, a thermocouple was placed into the catalyst chamber. The following equations were implemented to calculate the $CH_4$ and $CO_2$ conversions, $H_2$ and CO selectivity, and the ratio of synthesis gas ($H_2$/CO) (Equations (21)–(25)).

$$CH_4 \text{ Conversion } \% \;=\; \frac{(CH4)in - (CH4)out}{(CH4)in} * 100 \tag{21}$$

$$CO_2 \text{ Conversion } \% \;=\; \frac{(CO2)in - (CO2)out}{(CO2)in} * 100 \tag{22}$$

$$H_2 \text{ Selectivity } \% \;=\; \frac{(H2)}{2[(CH4)in - (CH4)out]} * 100 \tag{23}$$

$$CO \text{ Selectivity } \% \;=\; \frac{(CO)}{[(CH4)in - (CH4)out] + [(CO2)in - (CO2)out]} * 100 \tag{24}$$

$$H_2/CO \text{ ratio } \;=\; \frac{H2 \text{ Selectivity } \%}{CO \text{ Selectivity } \%} \tag{25}$$

## 4. Conclusions

The samples $Ni,Pd,Pt/Mg_{1-x}Ce^{4+}_{x}O$ and $Ni/Mg_{1-x}Ce^{4+}_{x}O$ (x = 0.00, 0.03, 0.07, and 0.15) (1% wt of Ni, Pd, and Pt loading) were developed utilizing the co-precipitation method with the precipitant $K_2CO_3$. Upon $CO_2$ reforming of methane reaction, the samples were utilized as catalysts for the syngas synthesis at a temperature of 900 °C, and $CH_4/CO_2$ of 1/1, with the $Ni,Pd,Pt/Mg_{0.85}Ce^{4+}_{0.15}O$ catalyst. The XRD analysis outcome indicated that traces of $CeO_2$ were found on both the MgO lattices and the surfaces of the catalysts. Results from this study demonstrated that some X-ray photoelectron signals were emitted from O1s, Mg2p, Ni2p, and Ce3d. The results of $H_2$-TPR showed that the $CeO_2$ reducibility was elevated with an elevation in the $CeO_2$ in the support with high active sites on the surface of the catalyst. For the DRM, a $CH_4$ and $CO_2$ conversion of 78% and 90% was recorded, respectively, with 1.1 $H_2$/CO ratio for the $Ni,Pd,Pt/Mg_{1-x}Ce^{4+}_{x}O$ catalyst. This result surpassed the catalyst $Ni/Mg_{1-x}Ce^{4+}_{x}O$ at 900 °C. Additionally, the $CO_2$ and $CH_4$ conversion for the $Ni/Mg_{1-x}Ce^{4+}_{x}O$ catalysts was reported to be lower than that for the $Ni,Pd,Pt/Mg_{1-x}Ce^{4+}_{x}O$ catalysts as a result of the presence of platinum and palladium metals that provided more electrons to nickel resulting in an enhanced electron density and inhibiting the catalyst coke sintering.

**Author Contributions:** Conceptualization, Y.H.T.-Y. and F.A.J.A.-D.; methodology, A.F.A.J., F.A.J.A.-D. and Y.H.T.-Y.; software, A.F.A.J and F.A.J.A.-D.; validation, A.F.A.J., Y.H.T.-Y. and F.A.J.A.-D.; formal analysis, A.F.A.J and F.A.J.A.-D.; investigation, A.F.A.J., F.A.J.A.-D.; resources, F.A.J.A.-D. and Y.H.T.-Y.; data curation, A.F.A.J. and F.A.J.A.-D.; writing-original draft preparation A.F.A.J. and F.A.J.A.-D.; writing-review and editing, F.A.J.A.-D.; visualization, A.F.A.J., F.A.J.A.-D. and Y.H.T.-Y.; supervision, F.A.J.A.-D. and Y.H.T.-Y.; project administration, F.A.J.A.-D. and Y.H.T.-Y.; funding acquisition, F.A.J.A.-D. and Y.H.T.-Y. All authors have read and agreed to the published version of the manuscript.

**Funding:** The authors thank the NanoMite Grant (Vot. No.: 5526308) for funding the research.

**Acknowledgments:** The authors would like to thank the PUTRACAT lab for permission to use the lab.

**Conflicts of Interest:** The authors declare no conflict of interest.

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
