# Peer review of "Enhancement of CO2 Reforming of CH4 Reaction Using Ni,Pd,Pt/Mg1−xCex4+O and Ni/Mg1−xCex4+O Catalysts"

_catalysts, doi:10.3390/catal10111240_

Round 1
Reviewer 1 Report
Al-Doghachi et al. reported in this work Ni and noble metals supported Ce-modified MgO catalysts for dry reforming of biogas. Despite the considerable number of prepared samples, the purpose is not clear and the analysis of their properties and performances is superficial. I cannot recommend this article for publication.
However, I think that authors should consider several aspects before an eventual new submission:
- Introduction section should be significantly improved and expanded in order to focus in the objective of the reported study. Additional studies where Ce was also used as promoter of MgO-containing biogas/methane dry reforming catalysts must be included, such as https://doi.org/10.1016/j.jcou.2017.12.015 or https://doi.org/10.1039/D0CY00039F. The novelty and innovative aspects of this work must be clearly identified;
- I recommend to add a table/scheme with the prepared samples in the experimental section for the sake of clarity. Also, and despite the use of similar catalytic tests conditions to other studies, it is crucial that this information appears in the manuscript;
- Tables and Figures presentation must be improved (e.g. properties presented in tables should be properly explained below the tables; attention must be payed to the use of several decimals in textural properties, do they have a meaning? which is the associated error?; H2 is not adsorbed in H2-TPR but consumed);
- A comparison with other catalysts for DRB/DRM from literature in terms of TOFs and H2 production rates would increase the impact of the work;
- Revision of English.
Author Response
Journal: Catalysts
Paper Title: Enhancement of dry reforming of methane reaction using Pt,Pd,Ni/Mg1- xCex4+O and Ni/Mg1-xCex4+O catalysts
Point to point answers to the reviewer’s comments
Reviewer 1
We thank the reviewer for the valuable comments. We have revised the manuscript according to the reviewer’s suggestions and provided a point to point answer to the reviewer’s comments.
C1. Introduction section should be significantly improved and expanded in order to focus in the objective of the reported study. Additional studies where Ce was also used as promoter of MgO-containing biogas/methane dry reforming catalysts must be included, such as https://doi.org/10.1016/j.jcou.2017.12.015 or https://doi.org/10.1039/D0CY00039F. The novelty and aspects of this work must be clearly identified;
Response: Done. The introduction section has been improved by adding the following paragraph: Akbari et al. [4] prepared Ni-MgO-Al2O3 nanocatalysts with various cerium contents by the co-precipitation and impregnation methods. Findings from the study demonstrated a high coke resistance and a high CO2 and CH4 conversion. In a study by Jin et al. [5], the catalytic performance of ALD-prepared Ni/Al2O3 catalyst was improved by the addition of CeO2 in which the addition of CeO2 resulted in a halt in the coke deposition and an increase in the stability of nickel-based catalysts.
The novelty of this work was in implementing new catalysts and new conditions for the DRM reaction. The important idea in the research was to reduce the carbon deposition to the lowest degree through several factors, the most important of which is the base support magnesium oxide, which adsorbs the carbon dioxide that combines with the precipitated carbon to produce carbon monoxide (equation 5). Secondly, we used a high calcination temperature to produce nanoparticles and reduce the coke formation. Another factor was the use of promoter in small quantities that contributed to reducing the carbon deposition according to the proposed mechanism. Also, the carbon deposition was reduced to minimum by implementing a stream of low oxygen (1.5%). On a final note, the use of three metals as main catalysts contributed to the activation of the reaction. Hence, all of these factors contributed to reducing the carbon deposition, and this was also proven in the study of Spent Catalyst whereby no clear effects of the catalyst were found after the reaction. It gave a slight increase in effectiveness.
C2. I recommend to add a table/scheme with the prepared samples in the experimental section for the sake of clarity. Also, and despite the use of similar catalytic tests conditions to other studies, it is crucial that this information appears in the manuscript;
Response: Added as recommended.
C3. • Tables and Figures presentation must be improved (e.g. properties presented in tables should be properly explained below the tables; attention must be payed to the use of several decimals in textural properties, do they have a meaning? which is the associated error? H2 is not adsorbed in H2-TPR but consumed);
Response: Done as recommended. I have included the requested data and figures (Fig.8-13). The decimals for some of the data have been removed as suggested.
C4. • A comparison with other catalysts for DRB/DRM from literature in terms of TOFs and H2-production rates would increase the impact of the work;
Response: Thank you very much for the very valuable suggestion. In my previous publications, I have never come across such a question. I have calculated the TOF based on the equation:
TON= moles of reactant consumed / mole of catalyst
TOF= TON / time of reaction (S-1)
H2 Conc.= 40 ppm= 40mg/L= 0.04 g/L
Weight of catalyst= 0.02g
M.wt of catalyst= 61.272 g/mol
And the obtained value was 4.08 *10-2. However, the value obtained in the literature was much higher (20-200). I am unsure if my calculation is right which is why I have not included the TOF. I look forward to your suggestion.
C5. • Revision of English.
Response: The English has been revised.

Reviewer 2 Report
Several methods are used to structurally characterize and catalytically evaluate Pt,Pd,Ni/Mg1-xCex4+O, and Ni/Mg1-xCex4+O catalysts in the dry reforming of methane. Most results are very interesting but their discussion needs clarification, particularly because of over-interpretation and blurred assertions. Several times the authors quote their own papers as if they were the first to work on DRM… It is strange is to have varied both Mg/Ce composition and metallic compositions (e.g. Pt,Pd,Ni/Mg0.97Ce0.03O and Pt,Pd,Ni/Mg0.93Ce0.07O) as if the contribution of metals and of supports to the catalytic properties was the same. The discussion of results is even more complicated.
The matter being enough, a strongly revised version should be proposed.
Examples are given below:
- First, Line 610, preparation of catalysts, there is no step of reduction before characterization and catalytic evaluation? Therefore during the catalytic testing the solids are in situ activated? This must be clarified.
- Line 109, what is this “catalyst complex Mg-Ce-O in cubic form”? Meaning of “complex”. Lines are for the support, not for the catalyst. Lines 119-120 sentences to be rewritten.
- Line 153, FTIR, “The unreduced catalysts spectra”, meaning of “unreduced”? The interesting part is at low wavenumber as far as the catalyst structural properties are concerned. The bands due to acetylacetonate (catalyst precursor?) could be mentioned and not shown. How is it possible to see bands of PtO and PdO for 1% loading?
- Lines 193 and foll. and Fig. 3: Why not showing Ce3d spectrum? Which are the species, Ce3+, Ce4+, Pt°, Pt2+ (etc.)? The surface composition is not reported, so it cannot be compared with XRF results.
- Lines 212 and foll., TEM, nor the “catalyst pores” nor the metallic particles are seen on the provided pictures. So results lines 222-225 are overinterpreted. We don’t see cubic particles but mostly hexagons which are characteristic of what? The scales are not readable on pictures.
- H2-TPR, lines 324-325: Meaning of sentence? How “MgCeO… interacted strongly with CeO2” and “resulted in a better Ni-O reduction” (meaning of “better”) and the remaining? Lines 347-349, meaning of sentence. Lines 365-367, how these TPR results provide informations about the catalytic activity (results provided further in the text)? This is not said.
- Line 393, TGA (in nitrogen), what the losses of weight stand for? Is it for catalyst or for acetylacetonate precursors? Does the reduction of Ce4+ happens? Is the Dm/m in accordance with such an hypothesis? Lines 398-400, sentence not understandable. On the contrary, the high melting points is not at all in favor of good interaction between MgO and CeO2 because their Tamman temperatures are still very high to allow wetting and possible interaction. More information would be obtained in hydrogen, that would allow comparison with H2-TPR.
- Catalytic results, line 429, what is the composition of the used “biogas”? Otherwise the term is fashionable but not true.
Lines 466 to 472 appears the terms “solid solution” for MgO-CeO2: where is the proof of such a formation for “the best catalyst Pt,Pd,Ni/Mg85Ce0.15O”? Not here in XRD results. See literature (e.g. J. Coll. Interf. Sci. 354 (2011) 341-342 where the demonstration is given that little Mg enters CeO2 as a solid solution, and not the reverse). - Lines 525 and foll., ref 10 is quoted (self quotation) but such reactions have been written a long time ago by other researchers!
- Post-reaction, lines 558-559, TPO-MS not shown. XPS analysis of carbon should be provided, that would be very informative, contrary to TEM.
FIGURES
- 1, the main lines of MgO and CeO2 should be shown by asterisks or others
- 2, FTIR, the abscissus is “wavenumber” and not “wave No”
- 6B, no ordinate; why is it so different from 6A?
- 7A the loss of weight between ca. 210 and 600 °C corresponds to what? Why Ni-supported is so different?
Increase or enhancement better than “elevation”
Author Response
Journal: Catalysts
Paper Title: Enhancement of dry reforming of methane reaction using Pt,Pd,Ni/Mg1-xCex4+O and Ni/Mg1-xCex4+O catalysts
Reviewer 2
We thank the reviewer for the valuable comments. We have revised the manuscript according to the reviewer’s suggestions and have provided a point to point reply to the reviewer’s comments.
C1. • First, Line 610, preparation of catalysts, there is no step of reduction before characterization and catalytic evaluation? Therefore, during the catalytic testing the solids are in situ activated? This must be clarified.
Response: Yes, the catalyst was reduced following the last step of the catalysts preparation, after which the catalyst was diagnosed with the required analysis. An exception was the FT-IR, which was analyzed before the reduction process to ensure the composition of the compound. For this reason, the final step (reduction of the catalysts) was not mentioned in the methodology.
Also, due to the lack of the usefulness of the FT-IR spectra in most literature when preparing catalyst, it was removed from the paper and the reduction step was added to the methodology.
C2. • Line 109, what is this “catalyst complex Mg-Ce-O in cubic form”? Meaning of “complex”. Lines are for the support, not for the catalyst. Lines 119-120 sentences to be rewritten.
Response: By complex, the authors meant the catalyst and not Mg-Ce-O support. Lines 119-120 were removed from the manuscript due to their redundancy.
C3. • Line 153, FTIR, “The unreduced catalysts spectra”, meaning of “unreduced”? The interesting part is at low wavenumber as far as the catalyst structural properties are concerned. The bands due to acetylacetonate (catalyst precursor?) could be mentioned and not shown. How is it possible to see bands of PtO and PdO for 1% loading?
Response: By unreduced, the authors meant the main catalyst in its ionic form (Ni2+) with acetylacetonate. Due to the presence of four compounds in the same figure, the figure was compressed and hence the bands were not clearly shown. In this case, the authors were aware of the need of adding another figure with magnitude and other figures for the reduced catalysts to clarify the figure. This is why the FTIR section was removed as it was not crucial in the manuscript. Also, most of the literature does not include the FTIR in the manuscript.
C4. • A comparison with other catalysts for DRB/DRM from literature in terms of TOFs and H2 -production rates would increase the impact of the work;
Response: Thank you very much for the very valuable suggestion. In my previous publications, I have never come across such a question. I have calculated the TOF based on the equation:
TON= moles of reactant consumed / mole of catalyst
TOF= TON / time of reaction (S-1)
H2 Conc.= 40 ppm= 40mg/L= 0.04 g/L
Weight of catalyst= 0.02g
M.wt of catalyst= 61.272 g/mole
And the obtained value was 4.08 *10-2. However, the value obtained in the literature was much higher (20-200). I am unsure if my calculation is right which is why I have not included the TOF. I look forward to your suggestion.
C5. • Lines 193 and foll. and Fig. 3: Why not showing Ce3d spectrum? Which are the species, Ce3+, Ce4+, Pt°, Pt2+ (etc.)? The surface composition is not reported, so it cannot be compared with XRF results.
Response: The Ce3d spectrum has been added to the manuscript. The spectrum has been conducted for the Ni/Mg0. 97Ce4+0.03O catalyst and not the Ni,Pd,Pt/Mg0. 97Ce4+0.03O catalyst and hence the catalyst does not include platinum. As for the graph, it represents Ce+4. Regarding the surface composition, the technician has not sent the composition for us to compare it with the XRF results and currently the machine is not working.
C6. • Lines 212 and foll., TEM, nor the “catalyst pores” nor the metallic particles are seen on the provided pictures. So results lines 222-225 are over-interpreted. We don’t see cubic particles but mostly hexagons which are characteristic of what? The scales are not readable on pictures.
Response: TEM provides the morphology of the catalyst (in the bulk) whereas SEM demonstrates the topology of the catalyst (catalyst surface). This means that the pores will only be shown when using the SEM and not TEM that are provided in figure (4). The metallic particles can be seen in Figure 4G.
Firstly, the XRD data were analyzed by the X-pert High score program that demonstrated a cubic form of the catalyst. This has also been supported by the TEM images that mostly appeared in the cubic form. Initially, the images were grouped and hence it was difficult to view the scale. However, the authors have now ungrouped the images to make the scales look clearer.
C7. • H2-TPR, lines 324-325: Meaning of sentence? How “MgCeO… interacted strongly with CeO2” and “resulted in a better Ni-O reduction” (meaning of “better”) and the remaining? Lines 347-349, meaning of sentence. Lines 365-367, how these TPR results provide informations about the catalytic activity (results provided further in the text)? This is not said.
Response: As for the lines 324-325, the explanation has been included in the manuscript. The lines 347-349 has been removed. Regarding the lines 365-367, from the TPR result, the H2-consumption with the highest value indicates a higher activity of the Pt,Pd,Ni/Mg0.85Ce4+0.15O catalyst when compared to other catalysts.
C8. • Line 393, TGA (in nitrogen), what the losses of weight stand for? Is it for catalyst or for acetylacetonate precursors? Does the reduction of Ce4+ happens? Is the Dm/m in accordance with such a hypothesis? Lines 398-400, sentence not understandable. On the contrary, the high melting points is not at all in favor of good interaction between MgO and CeO2 because their Tamman temperatures are still very high to allow wetting and possible interaction. More information would be obtained in hydrogen, that would allow comparison with H2-TPR.
Response: The loss of weight stands for moisture. TGA was conducted for the reduced catalysts and not acetylacetonate. No reduction of Ce+4 took place. The reduction of the catalyst was conducted before the TGA analysis. Most of the reviewed literature displayed findings for the TGA and not DTA.
Lines 398-400 were rewritten.
C9. • Catalytic results, line 429, what is the composition of the used “biogas”? Otherwise the term is fashionable but not true.
Lines 466 to 472 appears the terms “solid solution” for MgO-CeO2: where is the proof of such a formation for “the best catalyst Pt,Pd,Ni/Mg85Ce0.15O”? Not here in XRD results. See literature (e.g. J. Coll. Interf. Sci. 354 (2011) 341-342 where the demonstration is given that little Mg enters CeO2 as a solid solution, and not the reverse).
Response: Line 429 has been edited in the manuscript.
The question on lines 466 to 472 is worthy of attention. Whatever has been stated in the paper is true, but we implemented different reaction conditions in our research in which the samples were pressed into disks at 600 kg/m2 and then subjected to calcination at a temperature of 1150 ° C for a period of twenty hours to improve the texture, mechanical properties and the composition of the catalyst. This illustrates that the conditions have changed and the components also differed. Also, the used catalyst had a little amount of ceria which is much less than that of magnesia.
The suggested literature is excellent and has been added to the manuscript (reference 11).
C10. • Lines 525 and foll., ref 10 is quoted (self-quotation) but such reactions have been written a long time ago by other researchers!
Response: Done by changing the reference.
C11. • Post-reaction, lines 558-559, TPO-MS not shown. XPS analysis of carbon should be provided, that would be very informative, contrary to TEM.
Response: The TPO-MS has been added. As for the XPS analysis of carbon, the technician modified the data based on the carbon at 284.5 eV and we did not have carbon in the catalyst as it was reduced.
C12. • FIGURES
- 1, the main lines of MgO and CeO2 should be shown by asterisks or others
- 2, FTIR, the abscissas are “wavenumber” and not “wave No”
- 6B, no ordinate; why is it so different from 6A?
- 7A the loss of weight between ca. 210 and 600 °C corresponds to what? Why Ni-supported is so different?
Response: Figure.1- Done as suggested.
Figure.2- Removed along with the FTIR section
Figure.6B- Has been adjusted
Figure.7A- The weight loss corresponds to the loss of oxygen atom.
C13. • Increase or enhancement better than “elevation”
Response: Adjusted as recommended.

Reviewer 3 Report
see attached file

Author Response
Journal: Catalysts
Paper Title: Enhancement of dry reforming of methane reaction using Pt,Pd,Ni/Mg1-xCex4+O and Ni/Mg1-xCex4+O catalysts
Point to point answers to the reviewer’s comments
Reviewer 3
We thank the reviewer for the valuable comments. We have revised the manuscript according to the reviewer’s suggestions and have provided a point to point reply to the reviewer’s comments.
C1. • Line 24, without any comparison between the DRM results catalyzed by monoatomic Ni, monoatomic Pt, and monoatomic Pd, how to justify the tri-metallic Ni-Pt-Pd catalyst has the better performance?
Response: The subject of the current research is a comparison between mono-metallic Ni and trimetallic Ni-Pt-Pd. We will hopefully study palladium and platinum in the near future. The results of the mono-metallic Ni have been included to the manuscript (line 24).
C2. • Line 128, in table 1. For Ni catalysts, why Pt and Pd also appear in XRF analysis?
Response: Thank you very much for the comment and careful reviewing. We truly apologize for this typing error. The student who made the table had mistakenly overlapped the findings with another research.
C3. • Line 435, H2/CO ratio is 0.3%, how H2/CO ratio is defined?
Response: The ratio has been calculated based on eq. 24-26 in section 3.4. (Catalytic evaluations.
C4. • Line 449 in Figure 8, how the selectivity is defined and what values in Figure 8 corresponds to selectivity? Similarly, in Figures 9,10 and 11.
Response: The selectivity is defined by H2/CO ratio (syngas product) that is represented by the green bar in in figures 8-11.
C5. • Line 545 in Figure 11. In general, catalyst activity will decay as test time increases due to carbon deposition. In terms of reactant conversion or product yield, we should see decays of these quantities as test time increases. However, the variation trend of these quantities show a slightly increase as test time increases. Please explain.:
Response: This is very true. It is assumed that the catalytic activity decreases over time because of the deposition of carbon. However, the important idea in the research is to reduce the carbon deposition to the lowest degree through several factors, the most important of which is the base support magnesium oxide, which adsorbs the carbon dioxide that combines with the precipitated carbon to produce carbon monoxide (equation 5). Secondly, we used a high calcination temperature to produce nanoparticles and reduce the coke formation. Another factor was the use of promoter in small quantities that contributed to reducing the carbon deposition according to the proposed mechanism. Also, the carbon deposition was reduced to minimum by implementing a stream of low oxygen (1.5%). On a final note, the use of three metals as main catalysts contributed to the activation of the reaction. Hence, all of these factors contributed to reducing the carbon deposition, and this was also proven in the study of Spent Catalyst whereby no clear effects of the catalyst were found after the reaction. It gave a slight increase in effectiveness.

Round 2
Reviewer 1 Report
Dear Authors,
Thank you very much for the changes done in the article. The quality and scientific soundness of the presented data has been clearly improved.
Regarding the TOF calculations, I think there is a problem of concept. You can follow the formulas reported in the literature to calculate them (e.g. https://www.sciencedirect.com/science/article/pii/S092058611630298X). In my previous review report I suggested you to add a Table in the end of the manuscript comparing TOFs and H2 production rates from you samples and literature catalysts. Presenting both parameters and not only TOFs is relevant, as the second are highly influenced by the metallic dispersion and, also, these values are determined by attributing all the activity to the accesible active metal sites (limited when working with bi/multifunctional catalysts). Consequently, if authors do not have enough data for determining TOFs, I recommend to add a Table or Figure where H2 production rates (mol H2 produced per gram of catalyst and unit of time) determined for literature catalysts such as Ni/Al2O3, Ni/CeO2, Ni/Zeolites, etc at reaction temperatures where performances are far from equilibrium are presented.
Author Response
Journal: Catalysts
Paper Title: Enhancement of dry reforming of methane reaction using Pt,Pd,Ni/Mg1- xCex4+O and Ni/Mg1-xCex4+O catalysts
Point to point answers to the reviewer’s comments
Dear Reviewer 1
Special thanks for the valuable suggestions and recommendations to improve the quality of the manuscript. We have revised the manuscript according to the reviewer’s suggestions and answered the reviewer’s comments as can be seen below.
Q1) Regarding the TOF calculations, I think there is a problem of concept. You can follow the formulas reported in the literature to calculate them (e.g. https://www.sciencedirect.com/science/article/pii/S092058611630298X). In my previous review report I suggested you to add a Table in the end of the manuscript comparing TOFs and H2 production rates from your samples and literature catalysts. Presenting both parameters and not only TOFs is relevant, as the second are highly influenced by the metallic dispersion and, also, these values are determined by attributing all the activity to the accessible active metal sites (limited when working with bi/multifunctional catalysts). Consequently, if authors do not have enough data for determining TOFs, I recommend to add a Table or Figure where H2 production rates (mol H2 produced per gram of catalyst and unit of time) determined for literature catalysts such as Ni/Al2O3, Ni/CeO2, Ni/Zeolites, etc. at reaction temperatures where performances are far from equilibrium are presented.
Response: This indeed is one of the most important questions that I have come across and is of an excellent scientific value. It is the first time that I come across this question in the specialty of chemical engineering to show the efficiency of the catalyst in the interaction. Due to my chemistry background, this topic is new to me and following your suggestion, I have read a lot about it and came to the conclusion that what is meant by the catalyst is not the whole compound, but the main catalyst, which is the element of nickel in the prepared compound. Unfortunately, last time we have not measured the surface area of the cofactor. Hence, the calculations are incomplete and cannot be performed now in this experiment due to time constraints. Therefore, the subject and the incorporation of TOF will be taken into consideration in the future research.
For the meantime, we have included the H2 selectivity (Figure 12) as a simple way to address the issue. Thank you very much for the valuable comment and please accept our apologies for not being able to address it 100%.
Below are some of the literature that I have read:
- Kozuch, Sebastian, and Jan ML Martin. "“Turning over” definitions in catalytic cycles." (2012): 2787-2794.
- Goodwin Jr, JAMES G., Soo Kim, and WILLIAM D. Rhodes. "Turnover frequencies in metal catalysis: Meanings, functionalities and relationships." Catalysis17 (2007): 320-347.
- Alaba, Peter Adeniyi, et al. "Synthesis and application of hierarchical mesoporous HZSM-5 for biodiesel production from shea butter." Journal of the Taiwan Institute of Chemical Engineers59 (2016): 405-412.
- Boudart, M. "Turnover rates in heterogeneous catalysis." Chemical reviews3 (1995): 661-666.
As for the H2 production, my entire work is based on a comparison between two catalysts to demonstrate the role of adding Pt or Pt to the catalyst. Furthermore, the H2/CO ratio was also important to select the syngas in gasoline, methanol production etc.

Reviewer 2 Report
Please see attached doc.

Author Response
Journal: Catalysts
Paper Title: Enhancement of dry reforming of methane reaction using Pt,Pd,Ni/Mg1- xCex4+O and Ni/Mg1-xCex4+O catalysts
Point to point answers to the reviewer’s comments
Dear Reviewer 2
Thank you very much for the very careful review of the paper and the very valuable comments and information. It really helped us get a better perspective and have definitely helped improve the quality of the manuscript. We have revised the manuscript accordingly and answered the reviewer’s comments.
Q1) Lines 102-103, the XRD lines assigned to the “catalyst complex Mg-Ce-O in cubic form” cannot have (hkl) values if the structure has not been solved! Moreover, most lines are at the same 2Theta than either MgO or CeO2. Were there any lines of Ce2O3? If the term “complex” stands for Ni/Mg-Ce-O it must be written but it does not justify the assertion. Though there is very little amount of CeO2 its lines appear because Ce has a large cross-section. Same for Pd, Pt, probably the small lines at ca. 12-27° belong to them? Fig.1 must be revised accordingly. The A-b (red) pattern does not correspond to anything, may be an error? I would suppress it if no explanation. In B (caption, left), “CeO2” is written twice.
Response: *hkl has been removed. *True, although some of the peaks of ceria and magnesia appeared alike, however, the X-pert High Score software displayed double peaks; one for ceria and the other for magnesia. As for the Ni/Mg-Ce-O complex, *this is true which is why I have included ‘maybe’. *Figure 1 and Lines 102-106 have been edited.
Q2) Lines 129-130, XPS, line 129, “types of oxygen species” would refer to O22-, O-, O2-, only the latter is concerned. The O1s peak has several components and the written values (527 and 530) are not right. The peak could be deconvoluted using databank for XPS (CeO2 ca. 529-529.2 eV; MgO ca. 530 eV; NiO 529.1 eV; PdO 530.1 eV; PtO 529.8 eV).
Response: *Thank you very much for this valuable observation. As for the types of oxygen species, I did not mean the oxygen species oxide, peroxide, superoxide. What I meant was the bond of oxygen with Mg, Ni, and Ce. *The O1s peaks has been edited.
Q3) Lines 159-160, TEM. Due to the scale of TEM pictures (Fig.3), the mean size of Mg-Ce-O particles is roughly 100-400 nm and not what is written (“about 40-80 nm”). Lines 164-168, over-interpretation: how possible to see on these pictures that metals are “uniformly supported” … with “considerable homogeneity in metal dispersion”. Same over-interpretation for following lines, impossible to see these nanopores! Same for SEM pictures. The scale cannot be seen on provided pictures. So line 115 (before Table 1) must be edited.
Response: * The TEM scale has been removed. Lines 164-168 has been edited.
The scale cannot be seen due to the size minimization to fit the journal’s requirements. However, the scale can be clearly seen upon magnification. Line 115 has been edited.
The scale can be clearly viewed in the following FESEM and TEM images:
Q4) Lines 170 and foll., let me recall that the size of particles as determined by Scherrer’s formula stands for particles which are coherent to X-Rays (like single crystals). The real particles are always bigger as due to agglomeration, this size being determined by N2 adsorption or other techniques. Another reason to edit line 115.
Response: I have implemented Debye Scherrer’s to obtain the theta angle and beta (FWHM) from the highest XRD peak and not from all of them. The crystal size (D) was then calculated and the obtained value was lesser than 100 nm indicating that the catalyst was prepared as nanoparticles. The same method is utilized by most literatures. No agglomeration appeared in XRD. Line 115 has been edited.
Q5) Lines 208-209, as there are no adsorption-desorption isotherms and no pore size distribution provided nor explanations, one may suppose that the mean pore diameter was obtained by BJH method? Providing the figures, at least for the supports, would improve the quality of the paper. The caption to Table 2 says “fresh catalysts”, meaning?
Response: *The figures have been added. *By the fresh catalysts, I meant the catalysts used prior to the DRM reaction.
Q6) Lines 229 and foll., H2-TPR, it would have been informative to do the experiments on the bare supports (at least compare MgO and Mg0.85Ce0.15), because of the possible action of the metals. Let’s remember the SMSI effect. The discrepancy between A and B profiles in Fig.5 is not explained. For A, one recognizes the peaks of reduction of CeO2 (ca. 500 and 750 °C). Again, no scale of ordinates in B.
Response: *I have not conducted this experiment on Mg0.85Ce0.15O and no peaks were obtained when conducting the experiment on MgO alone. I only have the result for CeO2 and have included it in the TPR section. *The discrepancy between the two figures in which figure B had more peaks is due to the reduction of Pd and Pt that increases the H2 consumption for the tri-metallic The peaks of reduction of CeO2 were obtained by the report provided in the machine’s software. *The scales have been removed due to the presence of 4 peaks in the graph (most papers remove the scales when 4 or more peaks are present).
Q7) Lines 301-306, TGA, still no answer to my questions. Again a strong discrepancy between A and B. Line 302, the adsorption of N2 is unlikely (up to 2%!). But in N2 bottle there is a partial pressure of O2 ca. 10-5 that could be responsible for reoxidation of Ce3+ (now we know that the catalysts were reduced in H2, lines 529 and seq.). Calculating the weight loss for Ce4+ to Ce3+ gives ca. 1.4% (neglecting the amount of Ni). How to explain the total 6% loss in A and 10% in B? Line 307, ref [44] deals with Ni/CaO-TiO2-Al2O3, not appropriate to justify the TGA results.
Response: This is a great point. The nitrogen used in the thermal analysis instruments is of high purity. Although, through my theoretical observation, I realized that the only reason for the increase in the catalyst’s weight is N2 adsorption on the surface of the catalyst due to the high active sites in the catalyst.
As for the 6% loss in figure A and 10% in figure B, it is due to the loss of moisture. Also, figure B has Pd and Pt that can form coordination compound with water and hence increase the weight loss to 10%. In addition, this may be due to the loss of oxygen from the catalyst depending on the constituent’s interaction. Paper 27 have been changed.
Q8) Lines 356 and seq., the solid solution exists for a very little amount of Mg in CeO2, not the reverse (because the ionic radii are very different). There is no proof in this paper that it forms. Let me say that there is no need of “solid solution” to explain that adding a little amount of CeO2 to MgO modifies the properties of the Mg-Ce-O system. The explanation lies more in the presence of Ni (Pd, Pt) on, or close to, the boundaries between CeO2/MgO particles.
Response: The main question in my research is the role of the main catalyst Ni (Pd, Pt). However, by adding a small amount of CeO2 as promoter followed by the reduction of CeO2 to produce Ce metal on the surface of the catalyst also had some role in removing oxygen from CO2 that combines with carbon precipitating on Nickel’s surface and hence inhibits carbon formation (according to the suggested mechanism). As a result, CeO2 enhances the stability of the catalyst. This is based on what I understood from the question, I hope that answers your question.
Q9) Lines 521-522, about Ni/Mg-Ce-O preparation, a step of reduction by H2 was applied or not?
Response: Yes, reduction was applied and mentioned in line 561-562.
Q10) Other quotations not appropriate: Line 362, ref [30] (deals with solid solution NiO-MgO); Line 415, ref [33] (no CeO2).
Response: Sorry for this error. *Reference [30] has been changed. *As for ref [33], although promoter CeO2 was absent, the suggested mechanism can be used for most of the DRM promoters including the one used in our study.

Reviewer 3 Report
- In Figures 7, 8, and 9, why the horizontal axis is time (min)?
- H2/CO is just the ratio of H2 and CO, see your Eqs. (23) to (25).
- It is not clear why the experimental section is after the result discussion.
- In sec. 2.2.4. for the stability test, Fig. 10 demonstrated..... What are elevated methane rate and carbon oxide diffusion?
Author Response
Journal: Catalysts
Paper Title: Enhancement of dry reforming of methane reaction using Pt,Pd,Ni/Mg1- xCex4+O and Ni/Mg1-xCex4+O catalysts
Point to point answers to the reviewer’s comments
Dear Reviewer 3
We are really grateful the valuable comments and suggestions. We have revised the manuscript and answered the reviewer’s comments can be seen below.
Q1) In Figures 7, 8, and 9, why the horizontal axis is time (min)?
Response: Done. Edited as suggested.
Q2) H2/CO is just the ratio of H2 and CO, see your Eqs. (23) to (25).
Response: The H2/CO ratio is crucial for the step that follows the syngas production in which, if the ratio was 1, gasoline can be prepared in the Fischer-Tropsch reaction whereas if the ratio was 2, ethanol can be prepared and so on. I hope that this answers your question.
Q3) It is not clear why the experimental section is after the result discussion.
Response: This based on the journal’s requirements.
Q4) In sec. 2.2.4. for the stability test, Fig. 10 demonstrated.... What are elevated
methane rate and carbon oxide diffusion?
Response: This has been corrected in the manuscript.
